# Import options for chemical energy carriers from renewable sources to Germany

**Johannes Hampp** [1]*, **Michael Düren** [1], **Tom Brown** [2,3]

**1** Center for International Development and Environmental Research, Justus Liebig University Gießen, Hesse, Germany, **2** Department of Digital Transformation in Energy Systems, Technische Universität Berlin, Berlin, Germany, **3** Institute for Automation and Applied Informatics, Karlsruhe Institute of Technology, Karlsruhe, Baden-Württemberg, Germany

\* johannes.hampp@zeu.uni-giessen.de

## Abstract

Import and export of fossil energy carriers are cornerstones of energy systems world-wide. If energy systems are to become climate neutral and sustainable, fossil carriers need to be substituted with carbon neutral alternatives or electrified if possible. We investigate synthetic chemical energy carriers, hydrogen, methane, methanol, ammonia and Fischer-Tropsch fuels, produced using electricity from Renewable Energy Source (RES) as fossil substitutes. RES potentials are obtained from GIS-analysis and hourly resolved time-series are derived using reanalysis weather data. We model the sourcing of feedstock chemicals, synthesis and transport along nine different Energy Supply Chains to Germany and compare import options for seven locations around the world against each other and with domestically sourced alternatives on the basis of their respective cost per unit of hydrogen and energy delivered. We find that for each type of chemical energy carrier, there is an import option with lower costs compared to domestic production in Germany. No single exporting country or energy carrier has a unique cost advantage, since for each energy carrier and country there are cost-competitive alternatives. This allows exporter and infrastructure decisions to be made based on other criteria than energy and cost. The lowest cost means for importing of energy and hydrogen are by hydrogen pipeline from Denmark, Spain and Western Asia and Northern Africa starting at 36 EUR/MWh$_{LHV}$ to 42 EUR/MWh$_{LHV}$ or 1.0 EUR/kg$_{H2}$ to 1.3 EUR/kg$_{H2}$ (in 2050, assuming 5% p.a. capital cost). For complex energy carriers derived from hydrogen like methane, ammonia, methanol or Fischer-Tropsch fuels, imports from Argentina by ship to Germany are lower cost than closer exporters in the European Union or Western Asia and Northern Africa. For meeting hydrogen demand, direct hydrogen imports are more attractive than indirect routes using methane, methanol or ammonia imports and subsequent decomposition to hydrogen because of high capital investment costs and energetic losses of the indirect routes. We make our model and data available under open licenses for adaptation and reuse.

**Data Availability Statement:** The data underlying the results presented in the study are available from Zenodo at https://doi.org/10.5281/zenodo.7293102. The underlying model source code used for this study is available from GitHub at https://

github.com/euronion/trace/releases/tag/2022.11.
07 as well as the underlying technology cost
assumptions https://github.com/pypsa/technology-
data/tree/v0.4.0.

**Funding:** The author(s) received no specific
funding for this work.

**Competing interests:** The authors have declared
that no competing interests exist.

**Abbreviations: AE**, alkaline electrolysis; **AR**,
Argentina; **ASU**, air separation unit; **AU**, Australia;
**CAPEX**, Capital Expenditures; **CC**, carbon capture;
**CF**, Capacity Factor; **DAC**, Direct Air Capture; **DBT**,
dibenzyltoluene; **DE**, Germany; **DK**, Denmark; **EAC**,
Equivalent Annual Cost; **EG**, Egypt; **ES**, Spain; **ESC**,
Energy Supply Chain; **ESF**, energy surplus factor;
**EU**, European Union; **FOM**, Fixed Operation &
Maintenance; **FT fuel**, Fischer-Tropsch fuel; **FTD**,
Fischer-Tropsch-Diesel; **GDP**, gross domestic
product; **GEGIS**, GlobalEnergyGIS; **GIS**, geographic
information system; **GWA**, Global Wind Atlas;
**GWP**, global warming potential; **HVDC**, High-
Voltage Direct Current; **IRENA**, International
Renewable Energy Agency; **LCoE**, Levelised Cost of
Energy; **LCoH**, Levelised Cost of Hydrogen; **LHV**,
Lower Heating Value; **LNG**, liquefied natural gas;
**LOHC**, Liquid Organic Hydrogen Carrier; **MA**,
Morocco; **MCoE**, Marginal Cost of Energy; **MSR**,
methanol steam reforming; **OECD**, Organisation for
Economic Co-operation and Development; **PEM**,
proton exchange membrane; **PV**, photovoltaics;
**RES**, Renewable Energy Source; **SA**, Saudi Arabia;
**SMR**, steam methane reforming; **SNG**, synthetic
natural gas; **USA**, United States of America; **WACC**,
Weighted Average Cost of Capital; **WANA**, Western
Asia and Northern Africa.

## Introduction

Climate change mitigation efforts are driving energy transitions across the world. In these
efforts alternatives for established fossil energy carriers are being sought. These aspirations
gained additional traction with the plans of major international players like China, the Euro-
pean Union (EU) and United States of America (USA) to become climate neutral by the mid-
dle of this century. With technologies for the electricity sector already existing, these plans
require a shift of focus to the industrial, heating and mobility sectors. Today's and tomorrow's
energy demand of these sectors will have to be met with climate neutral and sustainable alter-
natives. The same requirements also hold for industrial feedstock which have to be de-fossil-
ised. For some countries producing chemical energy carriers and feedstock from domestic
RES or other near-zero-carbon energy sources may be an option. For other countries this will
prove challenging due to geographical, sociological or technological restrictions. Germany can
be considered such a country where limited potentials for domestic energy generation will pre-
sumably be insufficient to to meet energy demand for chemical energy carriers. Nowadays
Germany strongly relies on energy imports, which made up more than 76% (approximately
13.5 EJ) of Germany's domestically handled energy in 2018 [1]. Despite a high population den-
sity and mediocre RES potentials in a world-wide comparison, Germany has commited itself
to RES as a future source of energy. With these limitations the continued import of energy
should be investigated, where fossil carriers are substituted by synthetic chemical energy carri-
ers produced from RES.

Compared to electrical or heat energy, chemical energy carriers are easy to transport and
store, making them a preferred option for energy exports. Aided by pre-existing infrastructure
and experience from handling of fossil energy carriers, extensive use and significant trade vol-
umes of synthetic chemical energy carriers from RES can be expected by 2050. One option,
hydrogen, is currently receiving renewed world-wide attention with an increasing number of
nations adopting hydrogen strategies. A convergence to a system with one single predominant
chemical energy carrier might not be ideal: Depending on the end use the adaptation of pro-
cesses to a different chemical, e.g. hydrogen, will have to be weighed against substituting fossil
chemicals with synthetic drop-in alternatives. Adding to the complexity of this decision are the
different chemical and energy carrier specific properties which influence the conditions and
behaviour of a chemical during transport and storage. It therefore becomes important to gain
insight into the costs, composition and interaction of steps inside potential future chemical
Energy Supply Chains (ESCs).

Previous works have already analysed possible schemes for sourcing chemical energy carri-
ers. Fasihi et al. [2] conducted a world-wide analysis on how renewable energy sources may be
combined with other technologies to locally provide electricity and hydrogen at baseload qual-
ity and determined possible cost developments for 2020 to 2050. With a focus on synthetic
fuels another study by Fasihi et al. [3] analysed an ESC for Fischer-Tropsch-Diesel (FTD) and
synthetic natural gas (SNG) (methane) from the Maghreb region to Europe. Watanabe et al.
[4] and later Heuser et al. [5] modelled green liquefied hydrogen production and transport
from Patagonia to Japan. Ishimoto et al. [6] analysed the cost for transporting hydrogen and
ammonia from Norway to Japan and compared it with transport to the Port of Rotterdam in
Europe. Lanphen [7] also looked into ship-based supply chain options for importing liquid
hydrogen, ammonia and methylcyclohexane from various exporting ports to the Port of Rot-
terdam. Niermann et al. [8] explored the transport of hydrogen using a variety of Liquid
Organic Hydrogen Carriers (LOHCs) and compared their results against a hydrogen gas pipe-
line system to supply the energy carrier over a distance of 5000 km to Germany (DE). Schorn
et al. [9] compared shipping of methanol with shipping of $H_2$ (l) from Saudi Arabia (SA) to DE

and the economic viability depending on $H_2$ and $CO_2$ feedstock costs. More recently a number of global studies and analysis for importing hydrogen and other energy carriers from various regions around the world to DE have been released [10–13]. With a stronger focus on downstream infrastructure and energy distribution to end users, Runge et al. [14] compared the costs for transport fuels in a well-to-wheel analysis for mobility services in Germany. A global view on international hydrogen trade was taken by Heuser et al. [15] who modelled transport by pipeline and ship to determine optimal global supply costs. Later, International Renewable Energy Agency (IRENA) [16] gave an outlook on global energy trade scenarios for RES-based hydrogen and additionally ammonia.

While each case study provides important insights, it is difficult to compare these studies due to their different system boundaries, limited subsets of overlapping technologies, different energy carriers and regions investigated.

With this study we add several novel features to the existing literature. First we provide a comprehensive comparison of multiple ESCs from several different countries based on uniform assumptions and system boundaries. Secondly we deduct local electricity demand from the renewable resource availability in exporting countries, so that the best resources may be used locally. Thirdly we design our ESCs to work as islanded systems and be energy self-sufficient. Fourthly we make our data and model available under open licenses to allow for reproduction, adaptation and reuse.

## Materials and methods

In this section we first describe how our ESCs are structured. The investment optimisation problem is outlined followed by a description of the assumed technologies and a motivation for the countries selected. We then illustrate how RES potentials and feed-in are derived domestic demand is considered. We end this section by motivating our choice of Weighted Average Cost of Capital (WACC) and an overview of the technical model structure. The most important equations on which this model builds are given in S3 Appendix.

### Design of Energy Supply Chains

We model and investigate ESCs for chemical energy carriers starting at the energy source in an exporting country until the energy carriers are available in the importing country, in this analysis choosen to be DE. The ESCs considered in this study are export of a.) electricity by High-Voltage Direct Current (HVDC) with conversion to hydrogen in DE, b.) hydrogen gas by pipeline, c.) methane gas by pipeline, d.) liquid hydrogen by ship, e.) liquid methane by ship, f.) liquid ammonia by ship, g.) liquid methanol by ship, h.) hydrogen bound to LOHC dibenzyltoluene (DBT) [8] by ship, i.) liquid Fischer-Tropsch fuels (FT fuels) (kerosene-like) by ship. Fig 1 shows a schematic representation of all ESCs. For ESCs transporting ammonia, methane and methanol, an optional cracking step to hydrogen is further included for the case that the consumer needs pure hydrogen. Key properties of the chemical energy carriers are listed in Table 1.

The basic idea of the ESCs is shown in Fig 1. Detailed representations for all components, energy and chemical flows considered in each ESC are included in S1 Fig. In each ESC we consider

a.) sourcing of energy as electricity from RES,

b.) sourcing of the major chemical feedstock for synthesis i.e. water from seawater desalination, carbon-dioxide ($CO_2$) using Direct Air Capture (DAC) and nitrogen ($N_2$) using an air separation unit (ASU) from ambient air,

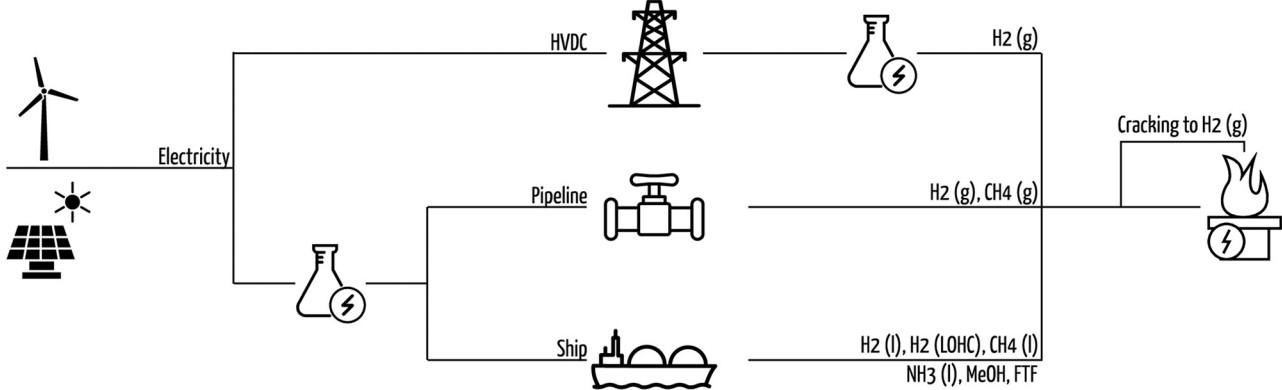

**Fig 1. Schematic representation of ESCs considered in this study.** ESCs cover electricity generation from RES, intermediary buffer storage for electricity and chemicals, conversion to chemical energy carriers, conditioning and transport from exporter to importer. Each ESCs delivers one out of five chemical energy carriers. Detailed representations of each ESC and all involved technologies are included in S1 Fig. License for icons: CC-BY-3.0 [17–21] and CC-0.

c.) electrolysis of hydrogen and optional synthesis (hydrogenation) of the chemical energy carrier,

d.) necessary conversion of the energy carriers for transport, e.g. compression or liquefaction,

e.) back-conversion into the energy carriers' usable form at ambient conditions, e.g. relaxation or evaporation.

If downstream processes require energy as input in addition to chemical feedstock, then this energy is provided from within the ESC. Either electricity is used for processes on the exporter side or the currently available chemical energy carrier is used, e.g. the propulsion energy for shipping is provided using the energy carrier transported by the ship and LOHC dehydrogenation uses part of the transported hydrogen. We exclude the transport within the exporting and importing country and storage in the importing country, i.e. excluded is

1. transport of electricity, feedstocks and chemical energy carriers between facilities and to the export terminal

**Table 1. Energy content of chemical energy carriers considered.**

| Energy carrier | State of matter (normal conditions) | Specific energy[a] [MWh/t] | Hydrogen content [wt. %] |
|---|---|---|---|
| Hydrogen $H_2$ (g) / (l) | Gas / Liquid (cryogenic) | 33.33 | 100 |
| LOHC (DBT)[b] | Liquid | 1.87[c] | 5.6[d] |
| Methane $CH_4$ (g) / (l) | Gas / Liquid (cryogenic) | 13.89 | 25 |
| Ammonia $NH_3$ | Gas / Liquid (cryogenic) | 5.17 | 17 |
| Methanol MeOH ($CH_3OH$) | Liquid | 5.54 | 12.5 |
| FT fuel | Liquid | 11.95 | - (var.) |

Values based on [22].

[a] All values represent the LHV.

[b] LOHC chemical is not consumed and reused, hydrogen is the chemical energy carrier delivered to the importer.

[c] $H_2$ share.

[d] DBT can hold up to 6.2% $H_2$, a depth-of-discharge for cycling at 90% is favourable for (de-)hydrogenation [14].

2. short-term or long-term storage at the importer in DE

3. distribution and end use of chemical energy carriers within DE

We exclude these steps because the range of possible options, such as the time patterns of the demand, would create too many scenarios and detract from the generality of our analysis. Such scenarios are better suited for investigation in specific case studies.

The ESCs are generally designed to not interact with any systems outside the ESCs. This design decision excludes the possibility for use of secondary products such as process heat and cooling services from cryogenic carriers, the sale of chemical by-products such as $O_2$. Also excluded are possible synergies by sector-coupling to heat and electricity systems on the exporter's and importer's sides, e.g. of industrial waste heat into the ESCs. Integration with other processes and commercial use of secondary products may provide grounds for business cases and lower market prices for chemical energy carriers.

## Investment optimisation problem

We model and optimise for least-cost investment of the essential components of ESCs to supply an annual energy demand. The ESCs components include electricity generators and generation based on historic weather data, conversion processes, buffer storage and transport between countries. The ESCs with their components, energy and mass flows are modelled using the open source modelling framework PyPSA [23]. We use a greenfield approach for our model and disregard existing infrastructure. We justify this modelling decision as a comparable scale of infrastructure described by the ESCs does not yet exist anywhere. Minimum annual investment costs are determined for each ESC based on the annualised cost $c_i$ of every single component $i$. The annualised cost represent cost for investment Capital Expenditures (CAPEX) $C_i$ and Fixed Operation & Maintenance (FOM) and are annualised using the Equivalent Annual Cost (EAC) method:

$$c_i = C_i \cdot (A_i + FOM_i) \tag{1}$$

where the component-specific annuity factor $A_i$ is

$$A_i = \frac{(1+r)^{\text{lifetime}(i)} \cdot r}{(1+r)^{\text{lifetime}(i)} - 1} \tag{2}$$

The annual interest rate $r$ is assumed equal to the selected WACC discussed below. The objective function to be optimised for minimal investment is

$$\min\left\{ \sum_{i \in \text{Generators}} c_i \cdot G_i + \sum_{i \in \text{Converters}} c_i \cdot F_i + \sum_{i \in \text{Storage}} c_i \cdot H_i \right\} \tag{3}$$

where generators (e.g. photovoltaics (PV)) are considered with their nominal capacities $G_i$ (e.g. MW), converters (e.g. compressors, electrolysers) with their throughput capacity $F_i$ (e.g. tonne per hour t/h, MW) and storage components (e.g. batteries, tanks) with their storage capacity $H_i$ (e.g. MWh, m$^3$). The optimisation is subject to constraints which ensure conservation of energy and mass flow and runs at hourly resolution. Most components may be freely dispatched without restrictions on ramping rates and minimum must-run capacities, as most technologies are usually flexible within a below hourly time-scale. Must-run capacities are only enforced for the synthesis of methane, ammonia, methanol and FT fuel, see the discussion in the technologies section. RES electricity feed-in is limited to the individual modelled time-series and excess electricity may be freely curtailed.

RES capacities may only be extended to the maximum potentials of their respective technology and resource quality class determined through geographic information system (GIS)-analysis, see the section on electricity supply below for details. The nominal capacities of all other components may be freely extended without limit. Capital costs for all components scale linearly with their capacity representing a situation where capacity expansion requires new facilities rather than extending existing ones. The used capital costs and FOM already assume large scale facilities with respective economies of scale applied as well as exogenous learning rates for cost reductions between 2030 to 2050. Process efficiencies are assumed constant for all years (see S4 Table) due to the difficulty of making well-founded estimates for future technological developments and improvements. One exception is made for hydrogen electrolysis where efficiency is expected to increase across all available technology options [24, 25], an improvement which would affect all ESCs presented here. Thus the electricity-to-hydrogen (Lower Heating Value (LHV)) efficiency for electrolysis is assumed to improve from 68% in 2030 to 71.5% in 2040 and 75% in 2050.

## Choice of technologies and energy carriers

The following section discusses the chemical energy carriers listed in Table 1 and their simplified production pathways. For most production pathways, alternative methods and integrated technologies with potential for efficiency improvement exist. Integrated technologies and alternative technology options are beyond the scope of our investigation and not discussed further.

**Renewable energy sources.**   We select utility PV, on-shore and off-shore wind as the only sources of energy for our model. We consider these technologies the only available ones with sufficient modularity and possibility for quick build up to high capacities while delivering near-zero carbon electricity necessary for a sustainable deployment. We consider the following other sources of energy unsuitable (with selected reasons):

a.) hydro power (prioritised for domestic demand, limited geographical locations, significant environmental impact, poor modularity and scalability),

b.) concentrated solar power (limited to specific geographical locations, low modularity and scalability),

c.) conventional nuclear fission (intransparent cost, unmodular and difficult deployment, sustainability issues with fuel and waste streams),

d.) nuclear fusion and unconventional nuclear fission (insufficient technology readiness level with unclear techno-economic prospects).

**Hydrogen.**   In all ESCs electricity is converted to the simplest chemical energy carrier, hydrogen, via alkaline electrolysis (AE). Compared to alternatives like proton exchange membrane (PEM) electrolysers, AEs electrolysers were historically considered to be less suitable to provide grid services due to their slow start-up times in the range of minutes [25]. For large scale operations in chemical energy carrier production the slow start-up time may be neglected because the electrolysis here does not need to provide grid services and the variability of RES feed-in can be managed using for example battery storage. The main advantages of AEs are that it is a well established technology which has in comparison to PEMs lower costs and does not require rare-earth-metals or platinum which may become bottlenecks for massive deployments in the future. In the case of large-scale hydrogen production and hydrogen-derived chemical energy carriers investigated here, we expect installations to run at high utilisation

factors with a large number of modular units that can combine into a large unit to create virtual flexibility, thus negating the main weakness of AE. The necessary water for electrolysis is produced through desalination of seawater via reverse osmosis to achieve the required purity and conserve existing, potentially scarce fresh water resources.

Hydrogen as a chemical energy carrier is versatile and may also be used as a feedstock for chemical synthesis. The physical properties of hydrogen make it difficult to transport as a gas due to its low energy density and as a cryogenic liquid due to the very low temperature required and high energy demand for liquefaction (0.203 MWh/MWh$_{LHV}$).

**Methane.**   Methane is an alternative gaseous energy carrier to hydrogen and is synthesised in established production processes through catalytic reaction of $CO_2$ with $H_2$ via the reverse water gas shift reaction:

$$CO_2 + 4\,H_2 \rightleftarrows CH_4 + 2\,H_2O \tag{4}$$

The synthesis is accompanied by adverse side reactions but can be run at high selectivity [26]. Transport of methane is well established and infrastructure for pipeline transport or transport as liquefied natural gas (LNG) already exists on large scales were synthetic methane may be used as drop-in replacement. Downsides are the emissions of $CO_2$ from combustion and methane during handling (leakage, slippage) and the high global warming potential (GWP) of methane. Liquefaction of methane requires less energy (0.036 MWh/MWh$_{LHV}$) than liquefaction of hydrogen and handling of cryogenic LNG is less complex and better established compared to liquefied hydrogen.

**Ammonia.**   Ammonia is another option for an gaseous energy carrier and was already used as energy carrier in past applications. Ammonia is synthesised by hydrogenation of nitrogen in the Haber-Bosch reaction:

$$N_2 + 3\,H_2 \rightleftarrows 2\,NH_3 \tag{5}$$

Feedstock nitrogen gas $N_2$ is available through extraction from the atmosphere via e.g. cryogenic ASUs. The ammonia synthesis process is well-established and an increasing focus in research for direct energetic applications of ammonia can be seen. In comparison to $H_2$ or $CH_4$, $NH_3$ has a lower specific energy but a high boiling point at $-33°C$, making it easier to handle as a liquid with a high volumetric energy density. In addition industry has long-standing experience of transporting and handling ammonia in its various forms, with around 10% of the global 183 Mt annual ammonia production being traded [27]. Direct energetic use of ammonia is possible but poses a range of challenges including suppression of $NO_x$ emissions [28]. Alternatively ammonia may be used as hydrogen carrier where the dehydrogenation of $NH_3$ happens through thermal decomposition. The dehydrogenation process has high energy requirements ($> 25\%$ of LHV [29]) and the technology is not yet fully commercially mature. Large specialised ammonia crackers are already in operation for the production of heavy water [27].

**Methanol (MeOH).**   Methanol (MeOH) is a liquid organic compound with favourable properties for transport, storage and energetic use. It is obtained by hydrogenation of $CO_2$:

$$2\,CO_2 + 6\,H_2 \rightleftarrows 2\,CH_3OH + 2\,H_2O \tag{6}$$

either in a direct catalytic methanolisation reaction or by an indirect route using a reverse water gas shift reactor to obtain syngas [30]. With an annual production volume of ca. 100 Mt [31] methanol synthesis is a well-established process with industry experience in handling, storage and transport. Methanol is used either as a industrial feedstock, for power and heat

generation, as a mobility fuel [32, 33] and also be considered a member the LOHC family with the dehydrogenated form being $CO_2$ [8]. Methanol-to-hydrogen cracking (steam methanol reforming) is mature on small industrial scales [34] and practical where methanol logistics are easier than logistics for alternative hydrogen feedstocks like e.g. natural gas.

**Fischer-Tropsch fuels (FT fuels).** Liquid fuels like Fischer-Tropsch-Diesel or kerosene can be produced from $CO_2$ and $H_2$ via Fischer-Tropsch synthesis. These fuels can be used as drop-in replacements for today's fossil based fuels and are simple to transport and store due to their liquid nature. The catalytic Fischer-Tropsch synthesis is not very selective and yields a mixture of hydrocarbon products, requiring post-processing to yield FT fuels [35]. We neglect gaseous outputs like fuel gases which constitute approximately 20% of the output [25] in our analysis and assume the products to have an average LHV of 11.95 $MWh_{th}$/t which is similar to regular diesel and aviation kerosene. Fischer-Tropsch synthesis is well established process using fossil syngas and infrastructure as well as experience from fossil hydrocarbon handling can be directly applied to FT fuels.

**Liquid Organic Hydrogen Carrier (LOHC).** LOHC is the last chemical energy carrier we consider which allows for piggyback transport of hydrogen. We choose DBT as a representative of the variety of LOHCs available [8]. Between its dehydrogenated ('unloaded', H0DBT) and hydrogenated ('loaded', H18DBT) form it may be loaded with up to 9 $H_2$:

$$9\,H_2 + C_{21}\,H_{20} \rightleftarrows C_{21}H_{38} \qquad (7)$$

For repetitive cycling and favourable (de-) hydrogenation a depth of discharge of 90% is more favourable [14] corresponding to 5.6 wt. % $H_2$. One significant advantage of DBT is that its properties do not change significantly between its hydrogenated and dehydrogenated form, allowing for the same infrastructure to be reused to achieve a closed LOHC cycle. The cost for the LOHC chemical DBT is assumed to be 2264 $EUR_{2015}$/t. The LOHC can be easily handled and stored with infrastructure similar to that of commonly traded liquid carbohydrates. To access the hydrogen stored it has to be dehydrogenated which requires about 28% of the hydrogen content as energy [8] as we assume the necessary heat has to be provided by the ESC itself and is not provided from an external source.

**$CO_2$ feedstock.** Methane, methanol and FT fuels require carbon dioxide ($CO_2$) as feedstock for synthesis. In our model all $CO_2$ feedstock is sourced from atmospheric $CO_2$ using DAC, thus creating a closed carbon cycle via the atmosphere between energy carrier synthesis and use. We do not consider the alternative approach of carbon capture (CC) at the location of use and back-transport of pure $CO_2$ to the location of energy carrier synthesis via a dedicated $CO_2$ infrastructure. This approach would require guaranteed capture of $CO_2$ from all use cases, which will prove complicated for applications where concentrated point sources are not available, like in aviation and individual mobility. Another obstacle to a perfect carbon cycle via CC is carbon leakage from imperfect CC which would need to be compensated through DAC infrastructure to ensure atmospheric carbon neutrality. Finally the infrastructure for recirculation of $CO_2$ would for most ESCs require additional dedicated infrastructure for $CO_2$ transport, incurring additional costs and complexity. While the costs of carbon capture at the location of use and back-transport may not be prohibitive, but such closed carbon cycles would require detailed analysis that is out of the scope of this paper. For the LOHC ESC recirculation is considered here as the infrastructure for recirculation of DBT in the LOHC ESC is the same as required for delivery of the hydrogen-loaded energy-carrying LOHC.

**Battery and chemical storage.** Storage technologies are essential for balancing the variable nature of RES electricity, buffering chemical feedstock and storing chemical energy carriers before export. Storage technologies smoothen the utilisation of downstream processes by

buffering variable upstream processes like RES or hydrogen production in RES-follow mode. Storage capacity expansion is an alternative way to increase process capacities by additionally increasing downstream utilisation rates and thus leading to lower Levelised Cost of Energy (LCoE) if the investment into storage capacity is lower than into process capacity expansion. In our model electricity from RES may be stored in a battery buffer storage. $CO_2$ as a feedstock gas may be stored in liquefied form. Hydrogen and methane may be stored for short term buffer storage in a compressed form as their liquefaction process is energy and capital intensive. Larger amounts of any chemical are only stored in liquid form to reduce the necessary storage volume. For hydrogen and methane this requires energy intensive and well insulated tanks to store both liquids at cryogenic temperatures. Underground storage like salt caverns for gases are not considered to keep our ESCs independent of location and geological conditions. In comparison ammonia liquefaction and storage is significantly easier as its boiling point is only $-33°C$ and thus ammonia may be stored in liquefied form. Storage tanks for methanol, LOHC and FT fuel are straightforward as they correspond to today's technologies used for light and heavy hydrocarbons. The storage technology options available for each ESC are as shown in S1 Fig. Storage capacities are endogenously determined by the model and represent optimal capacities under the given constraints minimising the objective function.

**Flexibility of synthesis processes.** In addition to the economic perspective of operating synthesis processes at a high utilisation rate, some synthesis processes may be designed for continuous operation from a chemical process point of view [25, 26] and not be suited for flexible operation or standby. We consider this by assuming a must-run capacity for the methanation and ammonia synthesis processes of 30% each, based on what could potentially be feasible for the methanation [26] and Haber-Bosch [25] processes. Methanol and FT fuel synthesis are assumed to run at a minimum of 94.25% capacity corresponding to a maximum of 3 weeks downtime for e.g. maintenance per year. The must-run capacity is assumed for the aggregated availability factor of the whole respective process plant, e.g. a must-run capacity of 94.25% translates to a maximum of 5.75% of the facilities capacities being unavailable for maintenance or other reasons at the same time (see S3 Appendix).

**Transport: Transmission lines, pipelines and ships.** Transfer of the energy between exporting and importing country plays a significant role due to the different characteristics between transport modes. HVDC transmission lines for transfer of electricity and pipelines for hydrogen and methane gas are already deployed technologies. These technologies allow for a continuous export-import supply stream. We consider here average costs, energy demand and distance-related losses for HVDC transformers, transmission lines, pipelines and pipeline compressors for above-ground if geographically possible. For imports from Argentina (AR) and Australia (AU) subsea connectors are unavoidable and we consider average subsea HVDC transmission line and subsea pipeline costs. The pipeline compressor costs in these few cases are the same as for above-ground pipelines due to a lack of reference numbers for long-distance deep-sea pipeline connectors. The numbers may thus prospectively underestimate real compressor costs. Shipping is the third class of transport modes considered and characterised by a non-continuous and delayed transfer of the energy carrier. We take this shipping characteristics into account by not allowing for concurrent use of loading and unloading infrastructure by different (groups of) ships at the same time. Rather than having ships travel at their (maximum) average cruise speeds and wait for the (un-) loading terminals to become available we create ex ante shipping schedules with lowered cruise speeds such that the arrival and departure of (groups of) ships does not overlap. The shipping distance along sea routes affect the shipping duration in the shipping schedules and the energy demands for propulsion, optional onboard refrigeration of boil-off gases and finally the necessary number of ships.

## Countries investigated

In our study we investigate large scale energy imports to Germany (DE) from the various countries shown in Fig 2. Germany as a country is interesting, as it heavily relies on energy imports today, is phasing out nuclear power and has less abundant renewable energy resources compared to other countries. We assume an energy import volume of 120 TWh based on the estimated hydrogen energy demand (LHV) for 2030 in the German Hydrogen Strategy [36]. On the one hand this number may seem ambitious given the high German fossil energy imports today and that its hydrogen production currently is mostly captive or merchant hydrogen from fossil sources without carbon capture. On the other hand we also presume an increase of this volume by 2040 and 2050 as not only fossil energy carriers but also industrial feedstock will have to be substituted by hydrogen or other chemicals. The import volume in our model is considered as annual demand since hydrogen uses and their demand patterns are yet to be known. This way it is decoupled from a specific demand pattern such with constant baseload or seasonally changing hydrogen demand and their case-specific buffer storage needs. As a reference we model and include chemical energy carriers produced from domestic resources in DE. The other countries included in our study are:

a.) Spain (ES), an EU country with high solar potentials,

b.) Denmark (DK), an EU country in close proximity to DE with high wind potentials,

c.) Morocco (MA), Egypt (EG) and Saudi Arabia (SA), representative countries in relative proximity to the EU with low population densities and high renewable potentials,

d.) Argentina (AR), a country repeatedly investigated by similar studies for exports of hydrogen to Japan,

e.) Australia (AU), a country discussed for energy exports to DE and as a possible future powerhouse for Asian countries.

The distance of an export-import route ESC influences the associated costs. With increasing distance investment costs, transportation energy demand, losses and duration (for shipping)

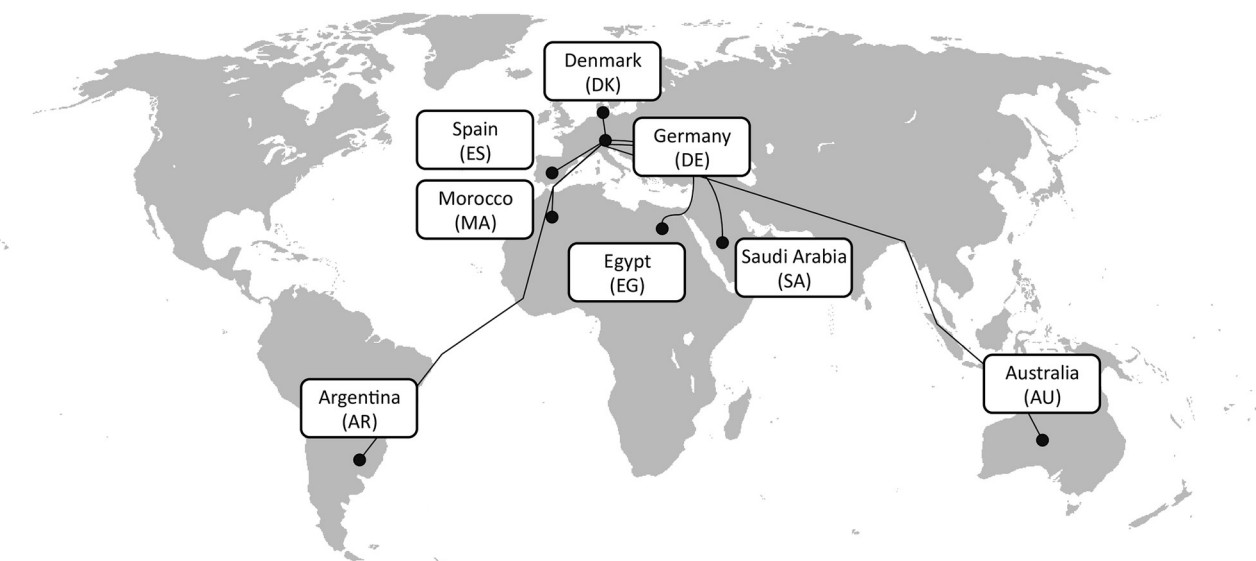

**Fig 2. Countries considered for export.** Exports of chemical energy carriers from countries shown to Germany (DE) are modelled and investigated for nine different Energy Supply Chains (ESCs).

**Table 2. Distances between exporting countries and DE assumed for each ESC.**

| From | Code[a] | Distance[b] [km] | HVDC line[c] [km] | Pipeline[d] [km] | Ship[e] [km] |
|---|---|---|---|---|---|
| Argentina | AR | 12 280 (3000) | 14 736 (3600) | 17192 (4200) | 13 056 |
| Australia | AU | 14 450 (3000) | 17 340 (3600) | 20 230 (4200) | 20 284 |
| Denmark | DK | 580 (0) | 696 (0) | 812 (0) | 812 |
| Egypt | EG | 3220 (0) | 3864 (0) | 4508 (0) | 6605 |
| Germany | DE | 0 (0) | 0 (0) | 0 (0) | 0 |
| Morocco | MA | 3330 (0) | 3996 (0) | 4662 (0) | 2938 |
| Saudi Arabia | SA | 4240 (0) | 5088 (0) | 5936 (0) | 12174 |
| Spain | ES | 1600 (0) | 1920 (0) | 2240 (0) | 3587 |

Bracketed values indicate the share of submarine distances considered.

[a] Based on ISO 3166–1 alpha-2.

[b] As-the-crow-flies distance between region centres [38], measured in Google Maps.

[c] Distance times a detour factor 1.2, own estimate.

[d] Distance times a detour factor 1.4, based on [39].

[e] Shortest sea route, determined with [37].

increase. The lengths of pipelines and transmission lines are based on the as-the-crow-flies distances between the country centres scaled by different detour factors for transmission lines (1.2) and pipelines (1.4). For AR and AU a land-only connection is not possible. We therefore estimate the share of distance to be traversed with submarine technologies based on the shortest cross-continental distances and scale them with the same detour factors as for land connections. A comparable approach with detour factors does not work for shipping routes as it does not account for land and water bodies. Instead we opt to use a freely available data source [37] to measure the shortest shipping routes. All resulting distances used are shown in Table 2.

## Electricity supply, demand and supply curves

For each country we model RES potentials and time-series based on results from a GIS-analysis and historical weather data using the GlobalEnergyGIS (GEGIS) model [40]. Eligible areas for PV and wind installations are determined on a 1 $km^2$ grid resolution by exclusion of protected areas, unsuitable land types and areas of high population density. Areas not within a 400 km radius of a gross domestic product (GDP) density of 100 000 USD/$km^2$ are further excluded where the threshold serves as a proxy to grid access and location accessibility. For a detailed description we refer to [40], for maps of the resulting eligible area see S4 Fig. Annual capacity factors are determined for all eligible grid cells and types of RES using ERA5 reanalysis weather data and data from the Global Wind Atlas (GWA) for 2013 as representative weather year. Grid cells are then categorised into one of 100 quality classes (0% to 100% Capacity Factor (CF)) for each RES technology based on their annual capacity factor. From the categorisation the potential of each quality class for each of the three RES technologies is calculated assuming a deployable potential of 1.45 MW/$km^2$ (PV) and 3 MW/$km^2$ (on-shore and off-shore wind). This potential is a compromise between technical potential, accessible potentials and social acceptance used in another study for the European electricity grid [41]. For PV these potentials may be conservative for less population dense regions and closer to the equator where others consider 75 MW/$km^2$ (PV) and 8.4 MW/$km^2$ (wind) feasible [2]. For large continuous wind farms power densities may need to artificially be reduced to lessen wake effect penalties [42]. These influences become more relevant for exporters with already high

potentials and flat supply curves and therefore should therefore not contribute a major influence on this studies' supply side.

In addition to the potentials we derive hourly generation time-series for each technology and quality class, resulting in a total of up to 300 independent RES time-series for each exporter.

From the annual generation for each quality class we can calculate each classes respective LCoE following

$$\text{LCoE(RES)} = \frac{\text{Annualised cost}}{\text{Annual generation}} = \frac{\text{CAPEX}}{G(2013)} \cdot \left[ \text{FOM} + \frac{r}{1 - (1 + r)^{-t}} \right] \qquad (8)$$

assuming technology specific parameters (Table 3), as well as $r$ = WACC and the modelled annual electricity generation $G$ in 2013. By ordering the class potentials based on their LCoE we obtain country specific electricity supply curves. Fig 3 shows examples using 10% p.a. WACC and technology (cost) assumptions for 2030.

Using the supply curve we account for projected domestic electricity demand: We generate electricity demand projections with a machine learning approach implemented by GEGIS [40]. The demand projections are based on global datasets for GDP, calendar days, temperature from ERA5 and the SSP2-34 'Middle of the Road' scenario [44] for 2050. This approach extrapolates past demand into the future and cannot account for structural changes like increasing demand through electrification. The projections are thus to be considered conservative estimates for electricity demand. The projected demands are shown in Table 4.

We consider the domestic electricity demand by removing the equivalent volume and RES with the lowest cost RES supply from our model. This corresponds to reserving the capacities with lowest expected LCoE for domestic use. The respective volumes are marked in the supply curves by the black dashed lines for the shown example 2030 and 10% p.a. WACC. Changes to the RES technology costs (year assumption) and WACC assumption affect the order of RES and therefore change RES with associated time-series available for export. Considering the shape of the supply curves in Fig 3, this approach noticeably affects the Marginal Costs of Energy (MCoEs) for DE and Denmark (DK).

## Choice of WACC

The choice of the costs of capital influences all cost calculations and is therefore crucial for meaningful results. [46] showed how configurations of a cost-optimised European electricity system change significantly from changes to WACC assumptions, especially non-homogeneous, country-specific assumptions. At the same time we are aware of the possible bias this might introduce, cf. [47].

**Table 3. Main technology assumptions for RES and electrolysis.**

| Technology 2030/2040/2050 | Lifetime [years] | CAPEX [EUR$_{2015}$/kW] | FOM [% p.a.] | Density [MW/km$^2$] | Efficiency [%] |
|---|---|---|---|---|---|
| PV (utility) | 40/40/40 | 376/329/302 | 1.9/2/2.1 | 1.45 | — |
| wind onshore | 30/30/30 | 1035/978/963 | 1.2/1.2/1.2 | 3 | — |
| wind offshore | 30/30/30 | 1573/1447/1416 | 2.3/2.3/2.3 | 3 | — |
| electrolysis | 30/32/35 | 450/330/250 | 2/2/2 | — | 68/71.5/75 |

Assumptions based on [25, 43] used for 2030, 2040 and 2050. CAPEX reflects the engineering, procurement and construction price. A full list of all technology assumptions is included in S2 Table.

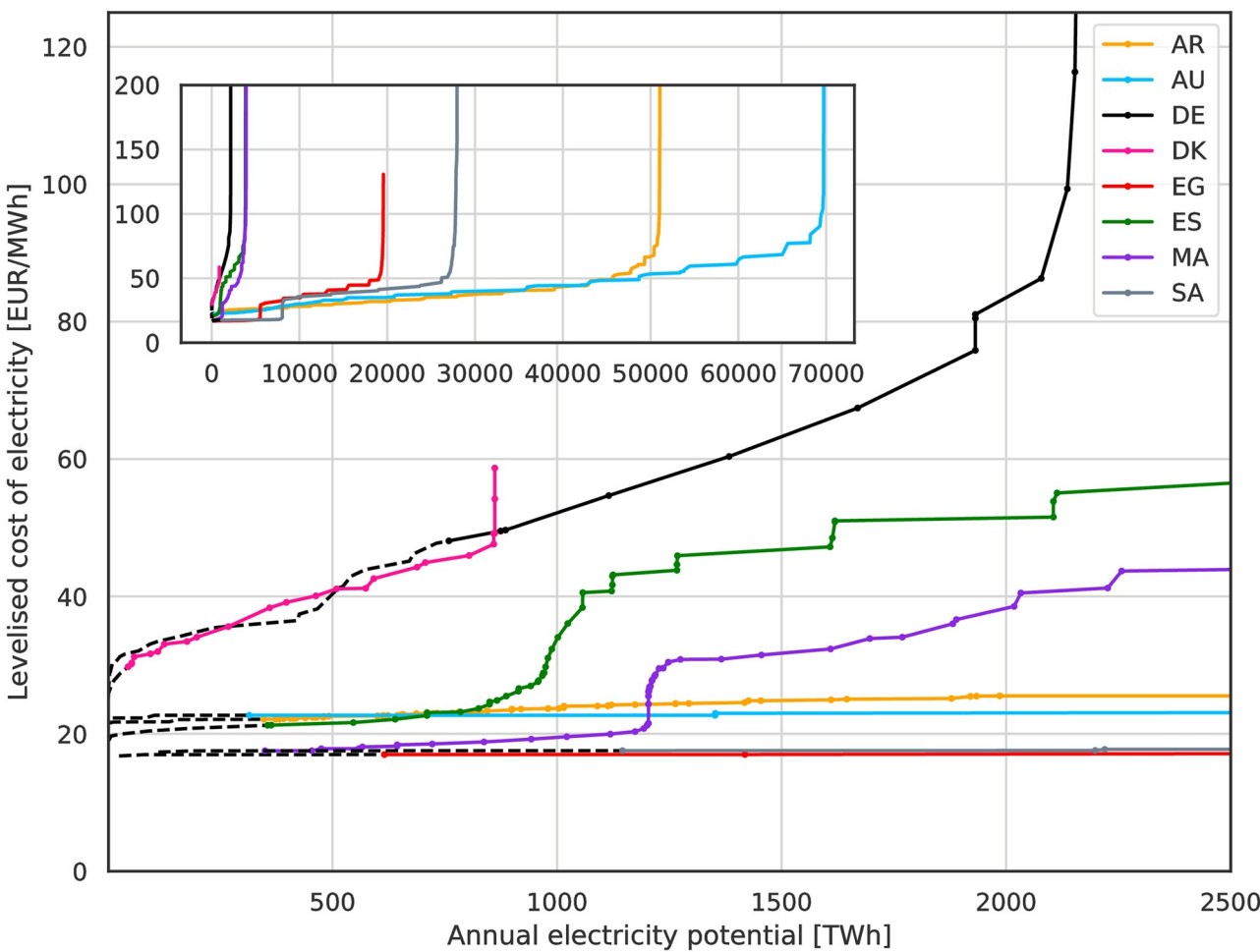

**Fig 3. Modelled electricity supply curves for 2030 at 10% p.a. WACC.** Dashed black parts are reserved for meeting domestic electricity demand and unavailable for export. The inlet contains the same plot on a larger scale. The visible step-wise increase in LCoE for ES and MA is where the cheapest electricity potentials from low cost PV are exhausted and the onshore and offshore wind enter the supply curve.

**Table 4. Used future (2050, projected) electricity demand and actual (2018) for reference.**

| Country | demand 2018 (actual)[a] [GWh] | demand used 2050 (projected) [GWh] |
|---|---|---|
| Argentina (AR) | 125 030 | 346 904 |
| Australia (AU) | 234 278 | 314 389 |
| Denmark (DK) | 32 865 | 44 854 |
| Egypt (EG) | 150 579 | 615 351 |
| Germany (DE) | 533 177 | 759 065 |
| Morocco (MA) | 29 678 | 122 419 |
| Saudi Arabia (SA) | 322 373 | 1 145 638 |
| Spain (ES) | 245 426 | 355 416 |

[a] Source: [45].

To retain comparability we assume time and technology independent WACC. We further choose to assume 10% p.a. WACC for all investments within all ESCs independent of the exporting country. This choice is founded on the assumptions used by IRENA, where the authors used inhomogeneous WACC assumptions of 10% p.a. for non-OECD countries and 7.5% p.a. for OECD countries and China in [48]. WACC for local RES projects usually depend on the technologies used and on individual project as well as country-specific risks [49]. It will therefore be interesting to see how WACC will develop for highly vertically integrated, multi-national and mixed-technology ESCs as presented here.

### Technical model structure

We use a multi-step workflow hard-linked using snakemake [50]. In the first part of the work-flow we utilise GEGIS [40] to determine potentials for RES, RES generation time-series and predict electricity demand. In the second part of the workflow we implement the ESCs in PyPSA [23] and combine them with the RES potentials, RES time-series and demand predictions into one dedicated PyPSA model for each combination of ESC and exporting country. The structure is also visualised in S2 Appendix.

## Results

Results are compared with a focus on their Levelised Cost of Energy and Levelised Cost of Hydrogen. The LCoE represent the costs for delivering 1 MWh of energy in the form of the energy carrier of the respective ESC, i.e. $H_2$ (g), $CH_4$ (g), $NH_3$ (g), methanol or FT fuel. For the Levelised Cost of Hydrogen (LCoH), costs are compared for delivering 1 MWh of $H_2$ (g) to the importer.

We first present LCoE for all ESCs and exporting countries for 2030. Then we show the development of LCoE based on technology cost projections up to 2050. Cost compositions for the ESCs are examined and main cost drivers discussed. We then continue by looking at the LCoH and by discussing sensitivities of the ESCs based on an sensitivity analysis for two selected scenarios. The sensitivity analysis shows an expected strong dependence to the choice of WACC on two selected ESCs from Spain (ES) to DE. We therefore also present LCoE and LCoH for 2030 to 2050 under a more optimistic choice for WACC of 5% p.a.. Additional results focusing on supply chain efficiency, curtailment rates and installed RES capacities are discussed in the S1 Appendix and S2 Fig. Presented results for LCoE and LCoH are included as tabular form in S1 and S2 Tables.

### Energy import costs for 2030 to 2050

LCoE, i.e. total system cost per $MWh_{th}$ delivered to DE, are shown in Fig 4 for 2030, 10% p.a. WACC and all exporting countries. The lowest cost options for import are by $H_2$ pipeline from DK at 75 EUR/$MWh_{LHV}$ and at 83 EUR/$MWh_{LHV}$ from Egypt (EG) and ES. All three ESCs take advantage of the low losses and investments associated with $H_2$ pipelines as static transport option and the short to medium transport distances to DE. With costs of 104 EUR/$MWh_{LHV}$ domestic $H_2$ production in DE is less attractive than these imports. $CH_4$ imports by pipeline are the least cost attractive option of the three (HVDC to $H_2$, $H_2$ & $CH_4$ pipeline) static transport connection ESCs. Methane in particular may be imported at lower costs as $CH_4$ (l) by ship rather than by $CH_4$ (g) pipeline. Cost performances of the shipping ESCs for $H_2$ (l), LOHC and $NH_3$ (l) are similar to those for $CH_4$ (l). They are followed by ship-based imports of methanol and finally imports of FT fuel by ship. There are some outliers for AR and AU for the static HVDC and pipeline connection ESCs. Reasons are the long transport

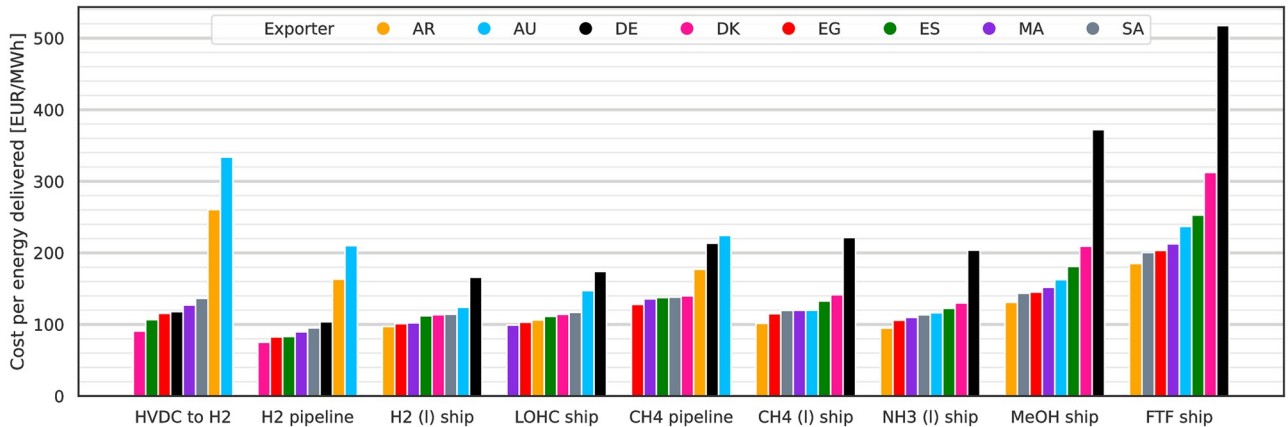

**Fig 4. LCoE in 2030 assuming 10% p.a. WACC by ESC and exporter.** LCoE are per MWh$_{th}$ delivered to DE. Lowest cost options are imports via HVDC with subsequent electrolysis in DE and H2 pipelines. For more complex and energy intensive ESCs on the right the order of preference is different compared to the static import options with imports from AR in all but one case being the cheapest.

distances for both exporters and the related high investment required for the HVDC and pipeline infrastructure.

An order of preference for the exporting countries can be identified: For statically connected ESCs short and mid-distanced exporters like DK, ES and EG are preferable. For ship-based ESCs, AR offers the lowest cost imports of energy to DE, making use of its good-quality RES, followed by imports from EG again. The least favourable position is taken by domestic production in DE: Generally for each ESC exists an alternative where the same energy carrier can be sourced for 75% of the cost of domestic production in DE. This development is driven by the low quality RES with high electricity costs and the approach we used to reserve RES capacities for domestic electricity demand. The approach reserved all PV potentials and only left wind resources for chemical production (cf. S2 Fig). Wind energy in Germany suffers from low output between June and September which the model compensates by over expanding wind capacities to keep supplying the synthesis processes. This is cheaper than using storage by batteries, which are not economical for storing more than a few hours worth of electricity demand, and there is no longer term electricity storage in the model. The remaining time of the year excess electricity is curtailed, causing high curtailment rates (cf. S1 Appendix) for DE. This leads to domestic production in DE to be competitive with imports only if H2 is produced without further processing. Under this condition the absence of need to transport chemicals internationally can compensate for the higher production cost in DE.

In Fig 5 the LCoE are shown declining in accordance with decreasing technology cost by 2050. Some of the projected LCoE decrease more strongly than others, most notable for methanol and FT fuel. While CH$_4$ (l) transport is cost competitive with CH$_4$ pipeline transport due to technological developments in the LNG industry over the past decades, for hydrogen the more complex of the transport chain and higher energy demands for liquefaction continue to make H$_2$ pipelines the preferred option for H$_2$ imports in the future. Noteworthy are the spreads between different ESCs and years. Neglecting domestic production in DE, the spread of LCoE for H$_2$ (l) or LOHC ship imports is low compared to the spread of methanol and FT fuel imports. It is also worth noting that the exporting country preference order does not change much between the years. Highest and lowest cost exporters stay the same and only some reordering in the cost mid-field can be seen where some countries benefit from anticipated cost developments more than others, see the mid-field options for methanol or NH$_3$

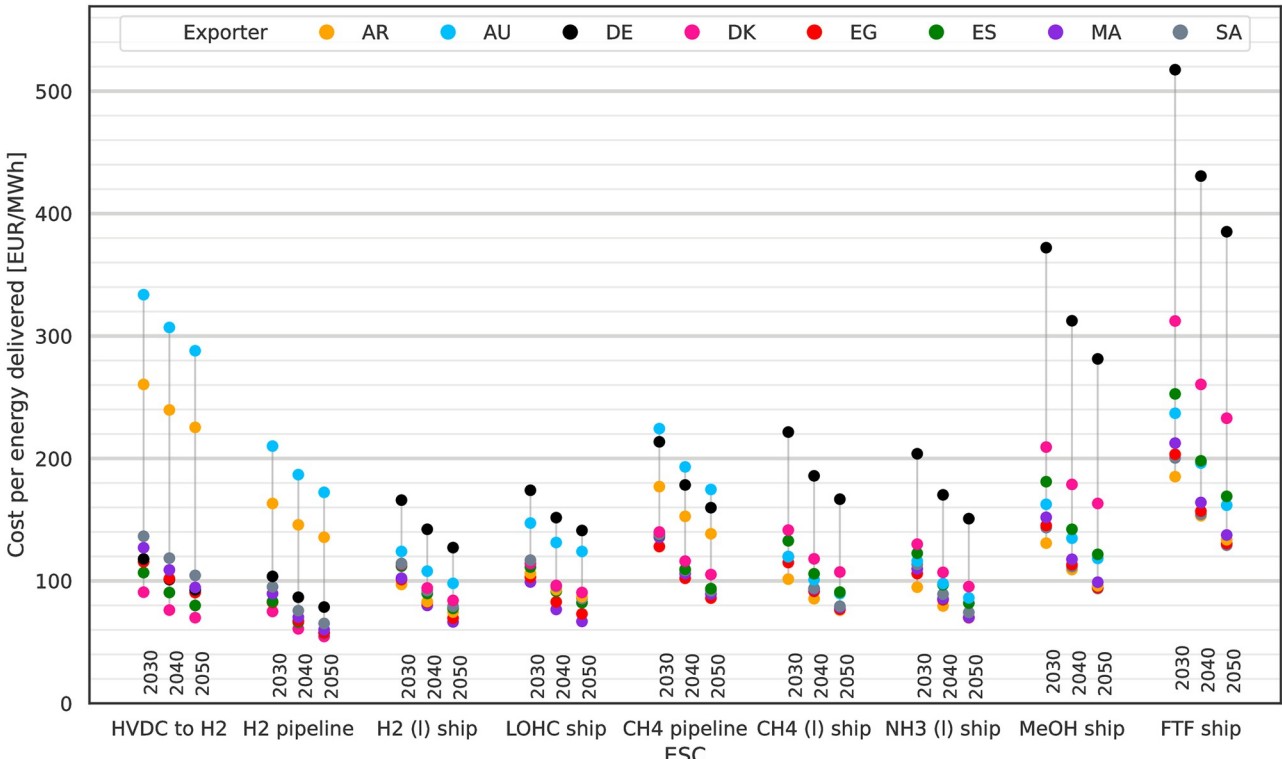

**Fig 5. LCoE in 2030 to 2050 assuming 10% p.a. WACC by ESC and exporter.** LCoE are per $MWh_{th}$ delivered to DE. The lowest cost import options are $H_2$ (g) imported by pipeline and electricity imports by HVDC with subsequent electrolysis. Methanol and FT fuel imports experience the strongest cost decrease linked to the anticipated cost reduction for synthesis and DAC.

shipping. This observation translates to no significant changes in the cost compositions of the ESCs, but rather just a general more or less homogeneous decrease of the total costs due to the cost technology reductions and electrolysis efficiency gains by 2050.

## Cost composition and drivers

Cost drivers can be individually identified for all ESCs by investigating their cost compositions. Fig 6 shows the cost compositions for three selected countries, comparing exports from AR and ES with domestic production in DE. Additionally a tenth ESC is included where ammonia is decomposed back into hydrogen for investigating hydrogen import costs discussed in the next section. The costs shown are the annualised cost per component. Energy costs are not attributed to the components and instead cause higher upstream investments for conversion steps or RES capacities to keep the ESCs self-sufficient.

RES generally make the single largest cost contribution between 39% to 55% (interquartile distance). The specific cost contribution depends on the ESC, exporter, the local energy demand and conversion processes involved. Electrolysis plays only a minor role with on average 5% of the costs in the shown cases. The costs for synthesis or liquefaction processes and DAC (for CO2-based ESCs) have a higher contribution than electrolysis. Related to the synthesis processes is the necessity for chemical feedstock storage required to operate the synthesis processes at and above their must-run capacities. The contribution from $H_2$ and $CO_2$ feedstock above-ground steel tank storage is higher for most ESCs than the contribution from electrolysis. Some feedstock storage is also deployed for domestic provisioning of $H_2$ (l) in DE but it

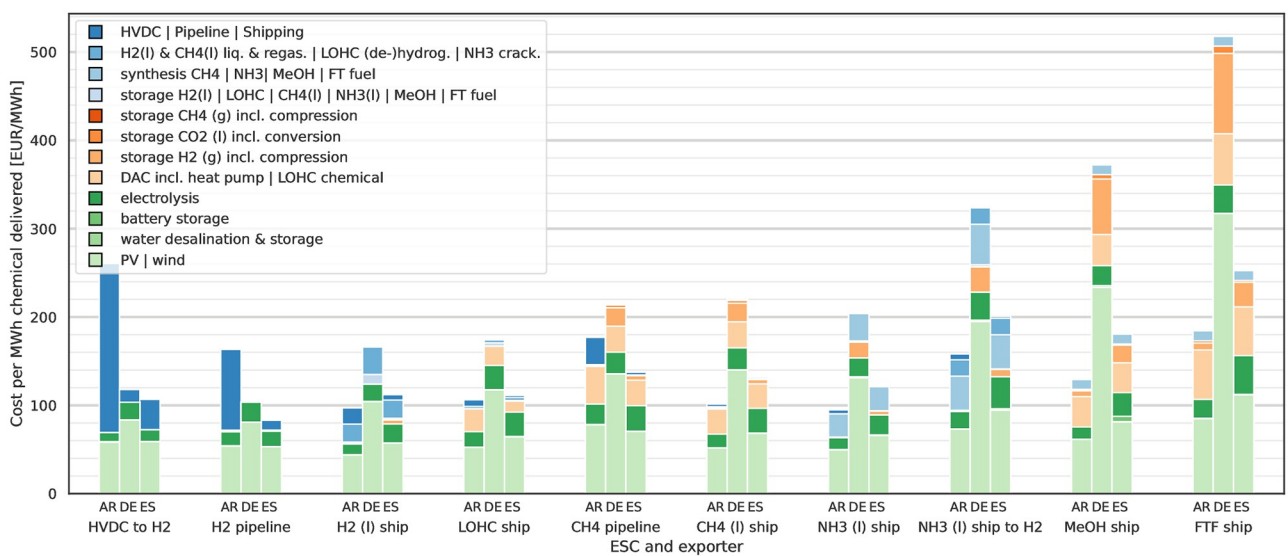

**Fig 6. Cost composition of selected ESCs for 2030 at 10% p.a. WACC.** Simpler, less energy intensive ESCs are favourable for short distances, e.g. domestic sourcing from DE and imports from ES, where low costs associated with transport can compensate high RES costs. More complex, energy intensive ESCs allow long distances exports, e.g. from AR, to become cost competitive due to lower RES costs and trading more complex molecules with higher energy intensity for lower transport cost.

cannot be seen for all H₂ (l) ESCs. The costs associated with shipping are negligible in comparison to HVDC and pipeline costs.

From Fig 6 we can also identify cost drivers for the bad performance of AR as an exporter with the static pipeline and HVDC to H₂ ESCs: The HVDC and H₂ pipeline connections have high investment costs due to the long distance between AR and DE. For the CH₄ pipeline connection the investment costs are only a third of the H₂ pipeline, but the LCoE is driven by energy demand of the CH₄ pipeline which is assumed to use the transported CH₄ (g) as energy source. This energy demand for transport requires additional capacities for CH₄ synthesis, H₂ electrolysis and CO₂ capture as well as RES capacities, thus making pipeline transport less attractive across medium and long distances compared to H₂ and CH₄ shipping. For the LOHC ESC the long shipping distance increases shipping time and thus makes larger volumes of the LOHC chemical necessary which is reflected in the share of cost of the LOHC chemical. In comparison to ES as an exporter the necessary LOHC chemical investment volume is 2 times higher for AR as exporter. Despite this the LCoE for imports from AR by LOHC are lower than from ES due to the country-specific differences in available RES: While the LOHC investment volume for ES is lower compared to AR, the LCoE in the electricity supply curve are higher for ES than for AR. For domestic production in DE the LCoE for electricity from RES are driving the ESCs LCoE even higher.

Furthermore the lower CF of RES in DE lead to higher capacities for electrolysis needed during peak production but with lower overall utilisation rate. The case of LOHC is the only ship-based ESC where imports from AR do not offer the lowest LCoE. Comparing the LOHC ESC and the other carbon-based shipping ESCs with the NH₃ (l) ESC shows the advantages of an energy carrier with low investment costs on the synthesis molecule (N₂) compared to the high cost for the LOHC chemical and CO₂ sourcing and handling. Inflexibilities in the synthesis processes do not directly become apparent from the cost composition figure but show up indirectly by the need for increased buffer storage capacities and overcapacities of components upstream in the ESCs, e.g. RES. A good example for this are the FT fuel and CH₄ (l) ESCs

where the more inflexible FT fuel synthesis leads to higher RES investments per MWh with higher curtailment and a different mix of RES capacities (see Fig 12 in S1 Appendix).

Battery storage is usually only deployed with limited capacities, able to sustain the ESC only for a few hours. In a few cases larger capacities of battery storage are deployed with a notable influence on the LCoE. Battery deployment can mainly be seen for HVDC, methanol and FT fuel-based ESCs, e.g. the HVDC to $H_2$ ESC for ES in 2040 (see S3 Fig). Investment into battery storage in the model coincides with higher shares of PV in the generation mix to increase the utilisation factor of downstream infrastructure, e.g. HVDC links, and to provide continuous electricity supply for must-run methanol and FT fuel synthesis processes. While sea water desalination is an important aspect for ensuring a sustainable production environment its costs and electricity needs do not contribute in a significant way to the final energy carrier cost.

Leverages for decreasing overall costs lie in reduction of storage costs by either direct reduction of the investment costs or usage of different technologies, e.g. cavern storage instead of steel-tank storage for $H_2$. An indirect leverage is the flexibility of synthesis processes as the must-run processes drive the storage volumes in our results, something which was shown for methanol by [51]. Increasing flexibility of processes and enabling lower must-run capacities as well as hot-standby would decrease the required storage volumes. Simultaneously it would increase the cost share of the synthesis process, such that cost improvements of the said processes could have a higher impact in lowering total product costs.

## Hydrogen import costs for 2030 to 2050

In addition to comparing the costs of energy delivered, we can also compare the ESCs based on their import costs for delivering hydrogen. For this we calculate the levelised cost of delivering 1 MWh $H_2$ (g) using adapted versions of the previously discussed ESCs. Comparing the cost of hydrogen rather than the cost of energy is useful for applications which either require pure hydrogen like hydrogen fuel cells or processes requiring hydrogen as an industrial feedstock.

The ESCs used are identical where the ESCs already delivered $H_2$ (g). The $CH_4$, $NH_3$ and MeOH ESCs are extended by cracking processes for converting the energy carrier to $H_2$ (g) with their respective energy demand and investment costs. The additional cracking processes are steam methane reforming (SMR), methanol steam reforming (MSR) and ammonia cracking. Additional water demand of the cracking processes is neglected. The extra process steps and their location in the ESCs are shown in S1 Fig. We exclude cracking of FT fuel as methanol can be considered an equivalent choice for the purpose of being a liquid, carbon-based hydrogen carrier under ambient conditions but with easier synthesis. FT fuels are better suited as drop-in fuel replacements.

Resulting LCoH for 2030 to 2050 are shown in Fig 7. There is a clear cost advantage for the four ESCs which import hydrogen directly (left side) compared to the four ESCs requiring an additional cracking step (right side). Lowest cost imports are from DK by $H_2$ (g) pipeline at 2.5 EUR/$kg_{H2}$ in 2030 and 1.8 EUR/$kg_{H2}$ in 2050. Other exporters located in close proximity, i.e. ES and the Western Asia and Northern Africa (WANA) countries, offer hydrogen at only slightly higher LCoH and can thus be considered comparable alternatives. For non-static imports via ship $H_2$ (l) and LOHC ESCs exports from AR and to some extent from AU are also attractive. By 2050 as most exporting countries show similar LCoH the question of exporter makes no big difference from a techno-economic point of view as there does not seem to be an inherent advantage to a specific exporter. For the remaining non-$H_2$-based ESCs the additional conversion steps required for $CO_2$-based and the $NH_3$ ESCs drive their energy demand

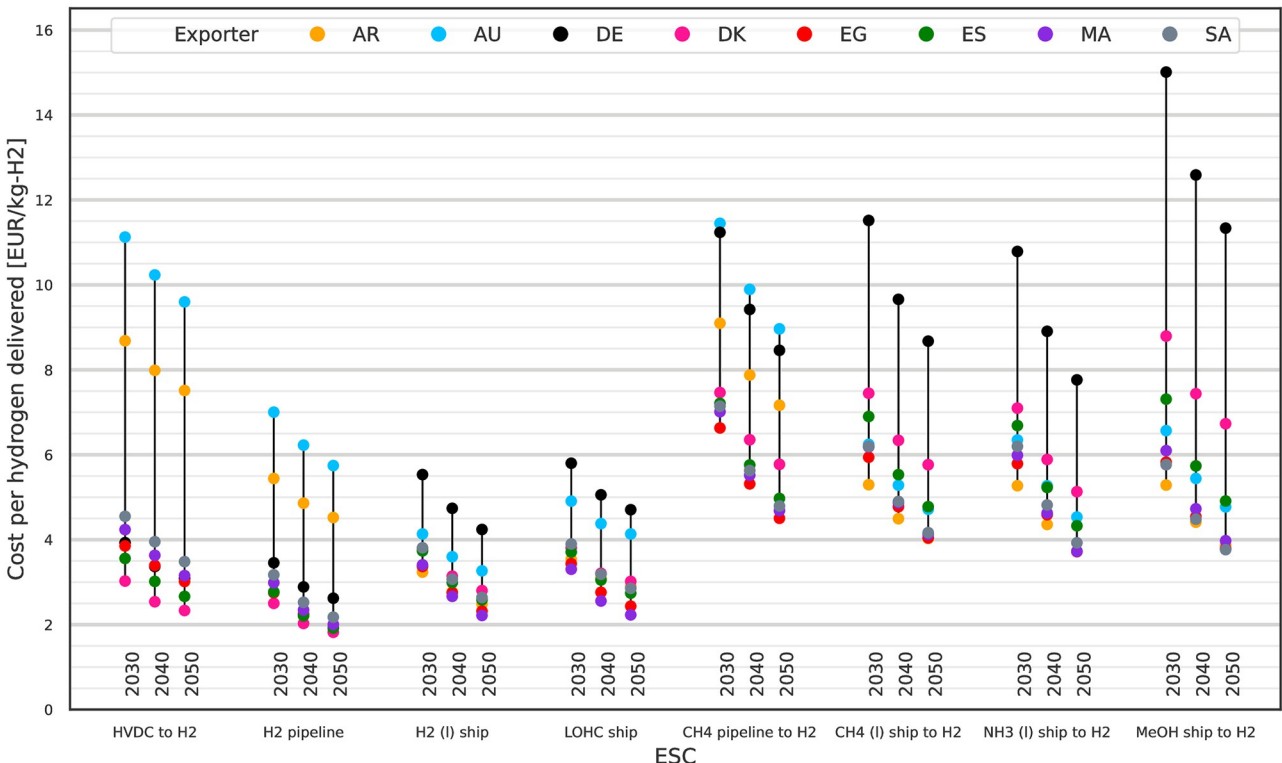

**Fig 7. Levelised Cost of Hydrogen (LCoH) for 2030 to 2050 using extended ESCs.** Direct hydrogen imports have lower LCoH than the alternative ESCs using methane, methanol or ammonia as energy carrier. The extended ESCs include extra process steps for cracking of the energy carrier to deliver hydrogen. All costs assuming 10% p.a. WACC.

and investment costs, leading to less attractive LCoH starting at 5.3 EUR/kg$_{H2}$ in 2030 decreasing to 3.7 EUR/kg$_{H2}$ in 2050 from various exporters via NH$_3$ and methanol.

If long-term storage were additionally taken into account, the ammonia and methanol ESCs would have an advantage over the CH$_4$-based ESCs due to easier long-term bulk storage. This advantage would translate into lower LCoH in comparison to the other ESCs with the advantage increasing with the storage duration. Examining the results for alternative exporter options, it shows that in general for each lowest cost option of an ESC an alternative exporter with similar or slightly higher LCoH exists. Neglecting the extreme outliers for AR, AU and DE, the spreads for pipeline based and direct hydrogen imports are lower than for ship based NH$_3$, CH$_4$ (l) and methanol imports. The higher spreads translate to a higher uncertainty of hydrogen import costs when trade relations change and a switch in exporting country become necessary. Choosing a chemical energy carrier and ESC for scale up based on lower spreads in LCoH and existing low cost exporter alternatives ensures the opportunity for increased competition and potential exporter substitution in the future.

## Sensitivity analysis

The sensitivities to exogenous parameter changes are visualised in Fig 8. Sensitivities are dependent on the scenario (year, ESC, exporter) by model design. We present quantified results for two selected scenarios of imports from ES to DE in 2030, by H$_2$ (g) pipeline and shipping of methanol. The exogenous parameters to be considered in the sensitivity analysis were pre-selected based on which parameters were expected to have the highest influence

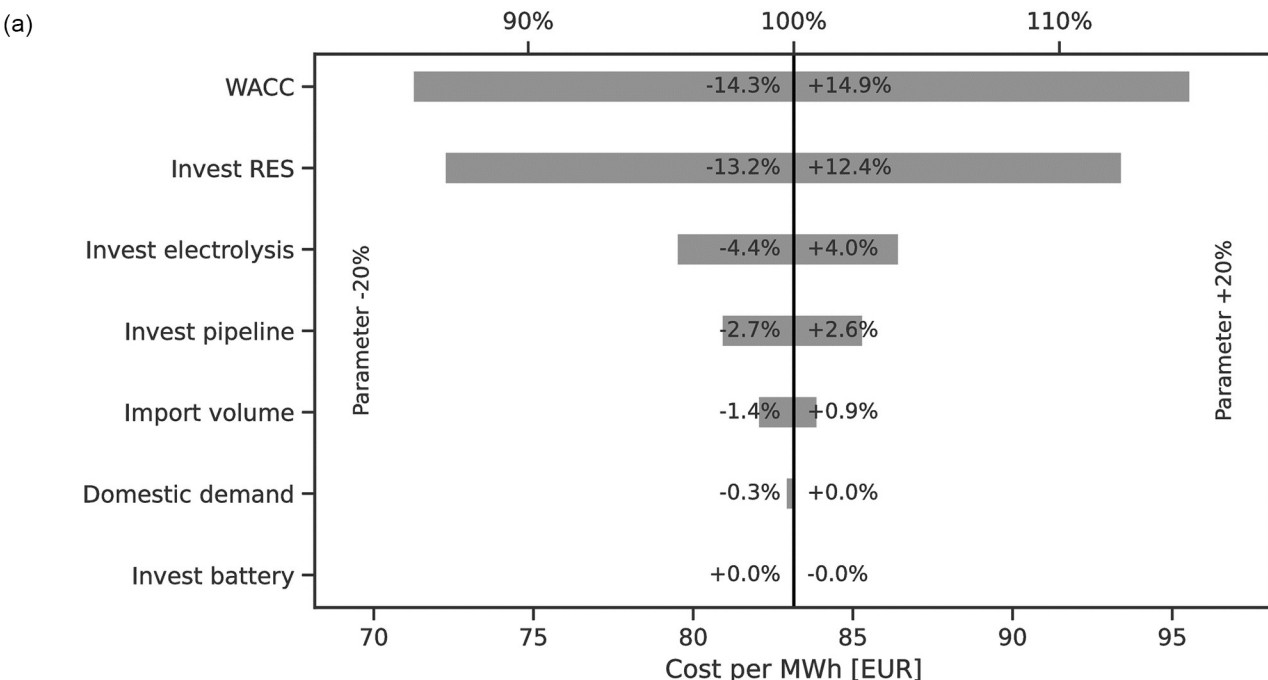

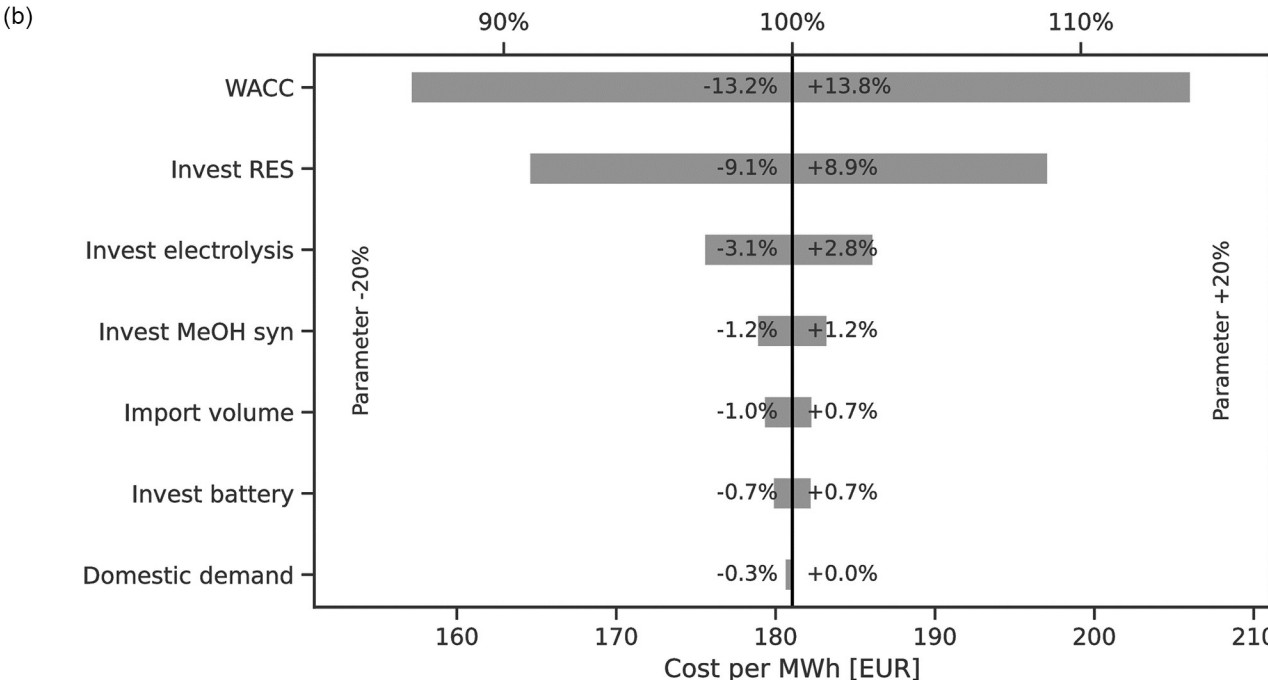

**Fig 8. Sensitivities for two ESCs (left) H₂ (g) pipeline and (right) methanol by ship from ES to DE for 2030.** Parameters listed on the left y-axis were varied by ±20%. The x-axis shows the resulting LCoE after variation and the relative change. The highest impact on the LCoE can be attributed to the choice of WACC followed by the cost of renewables and hydrogen electrolysis.

on the LCoE. For both ESCs one additional major cost contributor based on the cost composition analysis was added, i.e. CAPEX for $H_2$ pipeline including compressors and methanol synthesis.

The results show the highest influence on LCoE to be from the choice of WACC, followed by variations to CAPEX of RES and then the CAPEX of electrolysis or the ESC specific contributor (investment in pipeline or MeOH synthesis).

Reduction of CAPEX for batteries has no effect for the pipeline-based ESC and only a small symmetric effect on the methanol ESC. Variations of domestic demand and import volume are negligible. This is to be expected because the relevant supply curve range does not show considerable changes to the LCoE in the relevant area around 500 TWh of electricity demand.

## Lower WACC scenario

As indicated by the sensitivity analysis, the choice of WACC has the highest influence on the two ESCs analysed. Following this result we revisit the earlier analysis of LCoE and LCoH and rerun our model assuming 5% p.a. WACC (= −50%). Resulting LCoE are shown in Fig 9 and LCoH in Fig 10. The lower assumption is optimistic in comparison to other investigations like [48], but plausible and in line with recent reports like [52, 53] and in the face of similar or lower WACC reported for e.g. PV projects [49]. For large scale projects the assumption is also more reasonable as the projects include arguably high national interests and therefore support. The lower WACC leads to reductions of LCoE and LCoH each by around 35%. This exceeds most of the original scenario reductions seen between 2030 to 2050 due to technological learning. Under these more optimistic WACC assumptions one can find within each ESC one exporter and for each exporter one ESC with LCoE below 100 EUR/MWh$_{LHV}$. Similarly

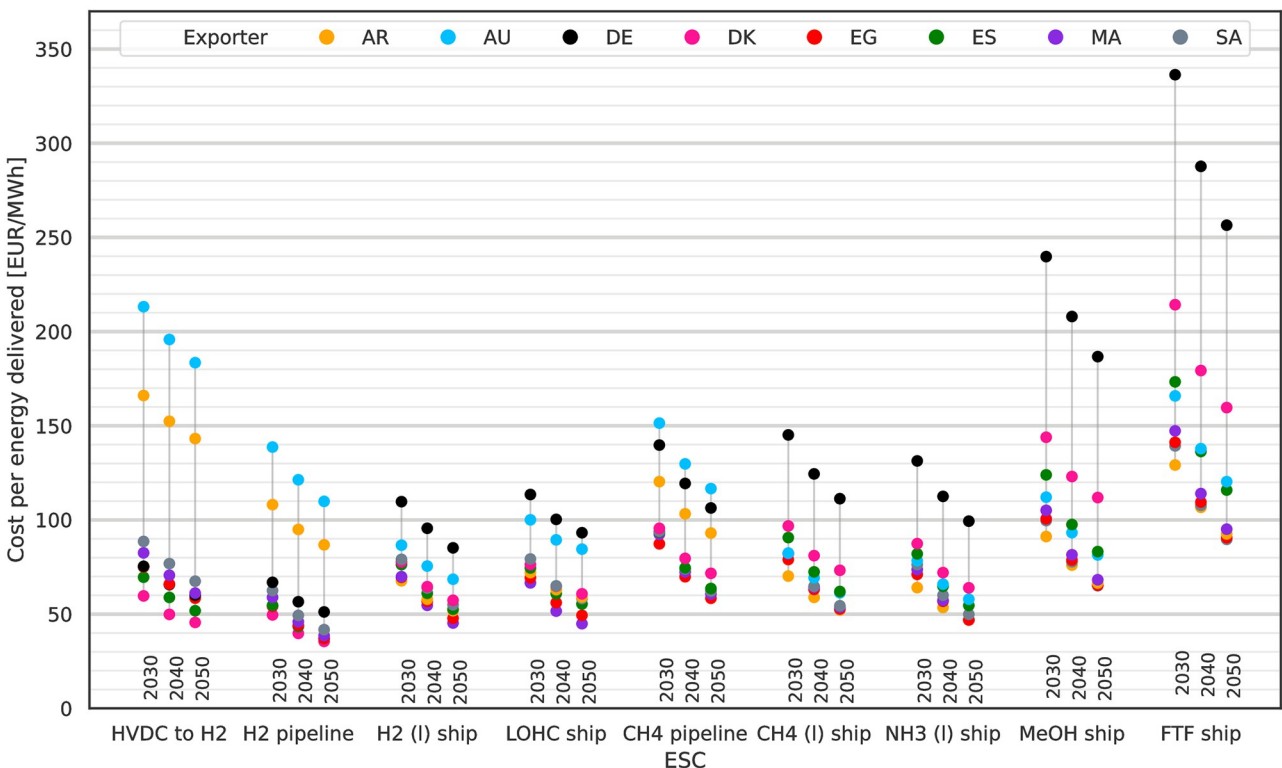

**Fig 9. LCoE for 2030 to 2050 at a reduced WACC of 5% p.a..** Lowering the WACC reduces LCoE by around 35%.

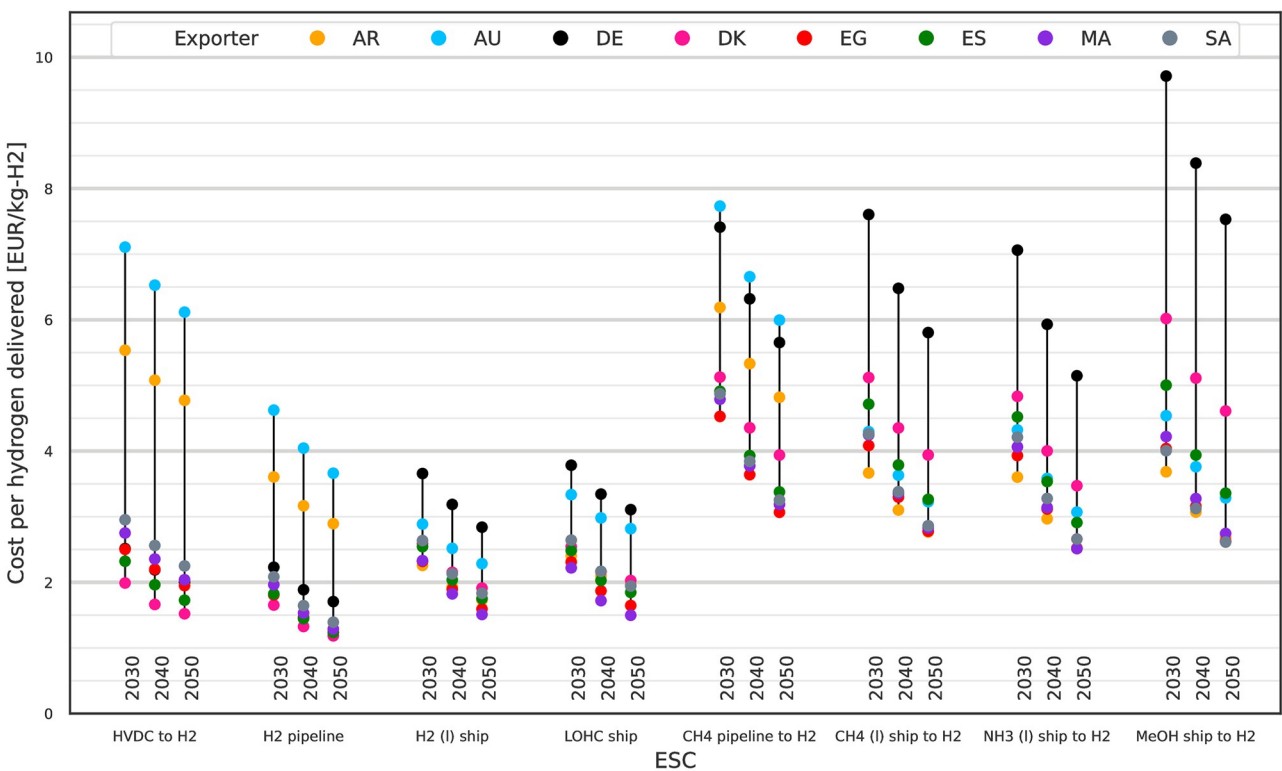

**Fig 10. LCoH for 2030 to 2050 at a reduced WACC of 5% p.a..** Lowering the WACC reduces LCoH by around 35%.

options for LCoH below 4 EUR/kg$_{H2}$ are available for all ESCs and exporters. What is left unchanged is the order of preference for exporting countries within each ESC; given that the change to WACC affects all exporters the same way this result is not surprising.

Lowest LCoE are 50 EUR/MWh$_{LHV}$ in 2030 to 36 EUR/MWh$_{LHV}$ in 2050 by H$_2$ pipeline from DK which are also the lowest LCoH at 1.7 EUR/kg$_{H2}$ in 2030 to 1.2 EUR/kg$_{H2}$ in 2050. By 2050 more than 20 ESCs offer options for imports of hydrogen at 2 EUR/kg$_{H2}$ (60 EUR/MWh$_{LHV}$) or lower, most of them being by a static transport connection via H$_2$ pipeline or HVDC but also including some shipping options.

## Comparison with today's commodity prices

We evaluate the competitiveness of our ESCs by comparing the future cost scenarios against today's market prices for the fossil-based counterpart commodities. The comparison between market prices and costs is not strictly valid because it ignores price formation in markets, but it can give a useful indication for the economic attractiveness of the ESCs. Fig 11 shows commodity prices relative to the average and median LCoE under 2050 technology assumptions with 5% p.a. and 10% p.a. WACC.

In our results methanol becomes available starting at 360 EUR/t from SA which is around 50% above 2020 market prices but within the range of market prices from 2021 and later. The current market prices are also within the range of the median methanol cost, indicating at least four exporter options for methanol within that cost range. Under less favourable financing conditions, cost differences exceed 100% and even the highest historical market prices are well below the 600 EUR/t median for methanol at 10% p.a. WACC. The constellation is similar for imports of FT fuel which hit break-even with the European average Diesel 2022 market price

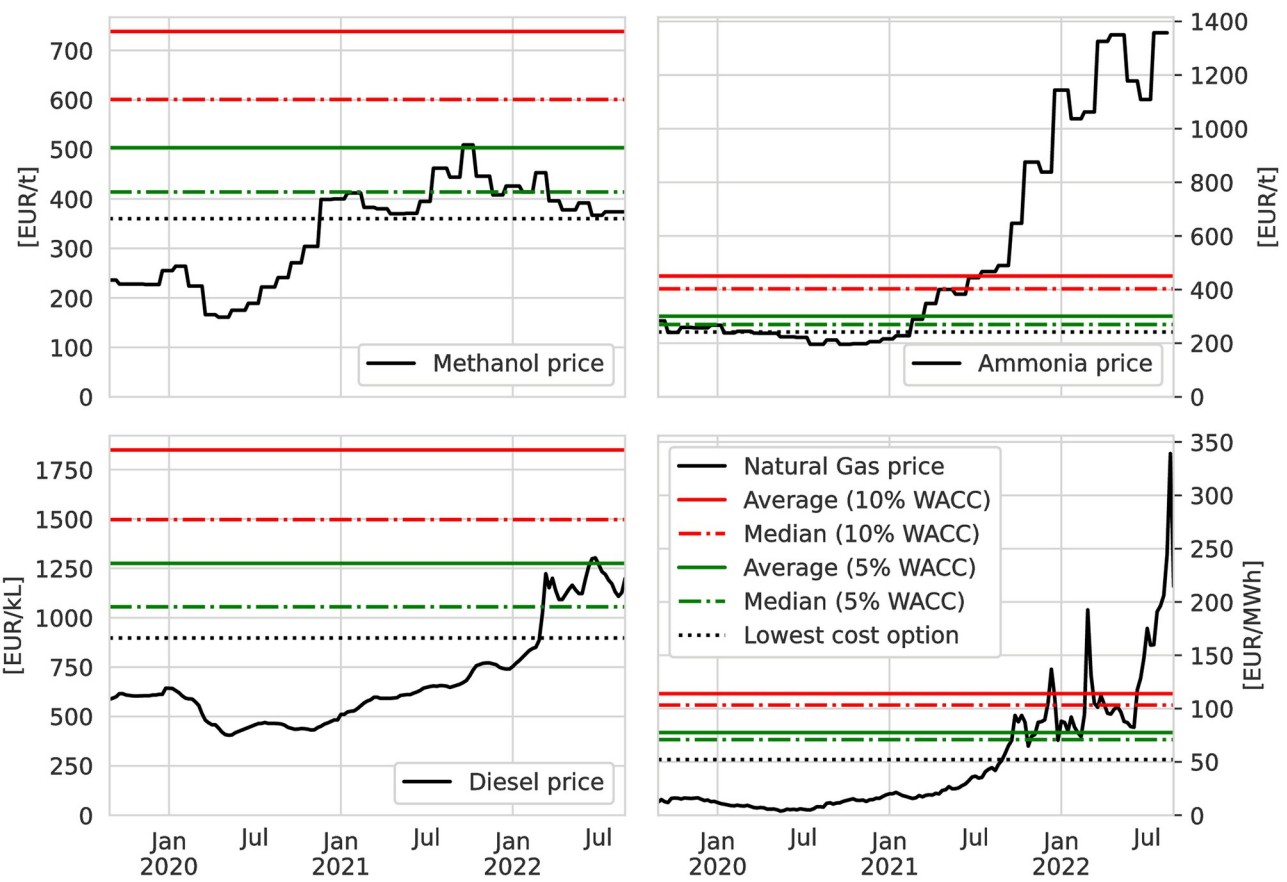

**Fig 11. Comparison of market prices of current fossil-based commodities with modelled costs for synthetic ESC-based alternatives.** Market prices are for Methanol based on MMSA Europe Spot FOB [31], Ammonia based on German export prices for ammonia [54], Natural gas based on Dutch TTF C1 future [55], FT fuel based on EU Diesel prices without taxes [56]. Statistics for natural gas are based on the costs for the $CH_4$ pipeline and shipping $CH_4$ (l) ESCs. The median value thus represents four exporter options for FT fuel, ammonia and methanol. For methane the median value represents eight exporter options.

under favourable WACC assumptions but are not attractive compared to 2021 and earlier market prices or under higher WACC.

Market prices of ammonia and natural gas are strongly correlated with natural gas, it being the major cost-driving feedstock for today's ammonia production. Until the middle of 2021 prices for natural gas and ammonia were moderate and about 50% below the costs of the lowest cost option available from our ESCs. In the second half of 2021 market prices started to increase and reached up to ten-fold of previous levels during the energy crisis in Europe. In the long-term market prices in Europe for natural gas and therefore also for ammonia can be expected to stay well above pre-2021 levels as pipeline-based gas imports are substituted with LNG. At higher market prices our modelled import costs are within the cost range of being attractive. Even median import costs for the less favourable 10% p.a. WACC scenario are below currently seen market prices.

The differences between costs and market prices can also be bridged through $CO_2$ prices on the direct emissions of fossil-based energy carriers, which may also make higher LCoE with unfavourable financing conditions (10% p.a. WACC) more attractive. Bridging e.g. a cost difference of 500 EUR/kl for diesel/FT fuel would be equivalent to a $CO_2$ price of 187 EUR/$t_{CO_2}$. A difference of 100 EUR/t on methanol could be compensated by a $CO_2$ price of 73 EUR/$t_{CO_2}$.

For natural gas a $CO_2$ price of 126 EUR/$t_{CO_2}$ could cover a difference of 25 EUR/$MWh_{LHV}$ which is the difference seen between mid July 2021 spot market prices and median LCoE from our scenarios. Such $CO_2$ prices are realistic when compared to the estimated $CO_2$ emission prices for 2030 of 129 EUR/$t_{CO_2}$ under stricter EU regulation [57] and are well within the range of the estimated social cost of carbon from $CO_2$ emissions in DE of 195 EUR/$t_{CO_2}$ to 250 EUR/$t_{CO_2}$ (2020 to 2050) [58].

This shows that under fast-track technology deployment that makes the 2050 cost projections appear earlier and under favourable financing conditions leading to low WACC, the differences in LCoE to the fossil chemical energy carrier alternatives could in some cases be overcome in the near future. It also shows the important role of enabling an environment of lower financing cost (WACC) to reduce the costs. Until the cost gap between synthetic RES-based and fossil energy carriers is closed, a purely cost-driven switch from fossil to synthetic fuels is improbable. Supporting policies, e.g. $CO_2$ prices and fuel quotas, or softer factors which incentivise change, e.g. adjusted customer preferences, exporter independence and diversification or business self-regulation following e.g. corporate social responsibility (CSR) or environmental social and corporate governance (ESG) criteria, will be required to drive the change.

## Discussion

### Limitations

The results we presented in this study are subject to a range of limitations. Some general limitations are due to the nature of the methods used while others arise from our model assumptions.

GIS-based analysis at low resolutions for determining RES potentials can over- or underestimate real potentials as it ignores local siting limitations below the resolution limit or not contained in the data input, e.g. terrain. The RES potentials and generation are likewise affected by using 2013 as representative weather year which disregards multi-year weather and climate change induced variations on RES generation and costs. By including exogenous technology development and learning for costs which use fixed projections for technology capacities being deployed in the future, we ignore the influence our ESCs capacities could have on world-wide learning and development. This limitation similarly extends to process efficiencies which are exogenously fixed and learning independent. For cost assumptions the validity and uncertainties cannot be assessed ex ante, a pitfall famously encountered by cost projections for RES technologies like PV which in hindsight were often too high [59]. Also some technologies might realise technological maturity faster or slower than others leading to earlier or later cost decreases in some ESC thus influencing the time-horizon of the results presented. There could also be path dependencies where existing infrastructure influences the choice of ESCs, such as existing LNG terminals leading to an advantage for SNG imports. Lastly assuming identical WACC for all countries and ESCs increases comparability but is not realistic; WACCs are generally country- and project-specific. More specific to our model we conducted investment optimisation for greenfield conditions in an islanded system. This assumption may not be appropriate where significant infrastructure for reuse or co-use already exists. Such cases are to be expected e.g. for electrical transmission lines and pipeline systems in the EU. Integrating energy exports with existing systems rather than deploying islanded systems can open opportunities e.g. for reuse of existing heat sources at lower costs than sourcing of the energy from inside an ESC. Reuse of existing infrastructure and integration both would lower the costs along an ESC. A similar effect should be expected from integration of technologies along the

ESCs, e.g. heat generated by electrolysis used for DAC, which was not considered either. Further we use simplified transport assumptions: Energy transport properties only scale by distance using representative average values which ignore e.g. topography, and we neglect shipping terminal costs. For forwarding electricity and chemicals between production locations and centralised facilities we assume copperplate-like transport without connection costs or losses on the exporter side and likewise for distribution systems on the importer side. On the demand side we presume an annual energy demand for domestic electricity as well as chemical energy carrier demand without specific demand pattern. This may cause unrealistically high amounts of low-cost, highly correlated electricity from PV to be used for domestic demand rather than some of its peak supply for the production of chemical energy carriers. The lack of demand pattern for the chemical energy carriers waives the need for short-term or long-term storage prior to end use where non-$H_2$ energy carriers may be better suited than any pure $H_2$ option. For the demand we further exclude differences in conversion efficiencies from energy carriers to energy services for better comparison. Potential competing demand by imports of other countries which may compete for the same ESCs is omitted as well as a to be expected increasing demand in DE exceeding 120 TWh. Aggregated demand may exceed available supply potential from some countries, e.g. DK, or regions, e.g. EU, cf. Fig 3. The decisions for transport, distribution and energy demand are expected to lead to an underestimation of LCoE. Another boost to LCoE can be expected as other nations start to import equally significant volumes of energy carriers, thus driving MCoEs in the supply curve and competition for best RES sites, unequally affecting LCoE from some exporters more, e.g. DK, than others, e.g. AU, AR. Attempting a generalisation for diverse distribution and end-use cases appears inadequate to us as it may lead to potentially misleading or easily misunderstood results. Instead we offer basic results on which further studies using sector or project specific parameters may be undertaken for which our results and model can act as a starting point.

In the broader picture we have considered imports to a single country, Germany, in our study. In reality there is to be expected a complex web of trade and competition between importing and exporting countries and individual actors. Our study is a first step into this direction.

## Comparison with other studies

Opportunities to compare our modelling results with other studies are limited to the various different system boundaries common in literature. [3] investigated import of FTD and SNG from the Maghreb region. For 2030 they arrived at 85.3 EUR/MWh$_{LHV}$ for FTD and for regasified SNG 98.8 EUR/MWh$_{LHV}$ assuming 5% p.a. WACC without $O_2$ sale benefits. Compared to our results with 5% p.a. (and 10% p.a.) WACC for imports from Morocco (MA) in 2030, their costs are lower than ours reaching 147 EUR/MWh$_{LHV}$ (213 EUR/MWh$_{LHV}$) for FT fuel and within a similar range for $CH_4$ (l) by ship at 82 EUR/MWh$_{LHV}$ (120 EUR/MWh$_{LHV}$). Differences can be found in [3] assuming DAC CAPEX to be a third of what we assumed and use a lifetime of 30 years compared to 20 years. They also assumed 37% lower CAPEX for AE, low cost cavern hydrogen storage rather than steel tanks, higher AE efficiencies and made use of heat integration into their process. Fasihi et al. [60] estimated 2050 costs for local ammonia production to be 260 EUR/t$_{NH3}$ to 300 EUR/t$_{NH3}$ for the best sites including AR, AU and the WANA region. For 'most habitable regions of the world' they estimate costs of around 450 EUR/t$_{NH3}$. The costs we arrived at in our study covers a similar range with costs (excluding DE) between 242 EUR/t$_{NH3}$ to 330 EUR/t$_{NH3}$ for the optimistic WACC case of 5% p.a. and 360 EUR/t$_{NH3}$ to 492 EUR/t$_{NH3}$ for the conservative WACC of 10% p.a.. The main differences between the studies are are the different assumptions on WACC with [60] assuming 7% p.a.

and their significantly lower CAPEX assumptions for hydrogen and battery storage. In terms of RES system compositions the authors saw a PV dominated system with complementary onshore wind deployed, while we see in our study quite different combinations of PV, offshore and onshore wind being deployed depending on the exporter and ESC.

Results from Niermann et al. [61] for hydrogen imports from Algeria by HVDC, ship with $H_2$ (l) and the LOHC DBT show significantly higher import costs at 6% p.a. WACC when compared with our results for MA in 2030 with 5% p.a. WACC: They report 11.5 EUR/$kg_{H_2}$ for LOHC shipping (we: 2.2 EUR/$kg_{H_2}$), 13.2 EUR/$kg_{H_2}$ for $H_2$ (l) shipping (we: 2.3 EUR/$kg_{H_2}$) and 15.6 EUR/$kg_{H_2}$ for HVDC imports (we: 2.8 EUR/$kg_{H_2}$) The stark cost difference is driven by their assumptions of a different electrolysis technology (PEM) with different technological parameters, fixed electricity costs of 50 EUR/$MWh_{el}$ and inclusion of seasonal storage as well as distribution costs, i.e. assumptions which are expected to drive costs for pure hydrogen storage (gaseous or liquid) and investment costs for the LOHC chemical substantially.

Lastly compared to [15] we see comparably flat supply curves for regions with large RES potentials. The lowest costs for $H_2$ (l) exports from Patagonia (AR) seen by Heuser et al. [15] are at 3.06 EUR/$kg_{H_2}$ excluding shipping cost. The authors further estimate the shipping costs to add another 0.53 EUR/$kg_{H_2}$. These results are significantly higher than our results for 2050 with 2.5 EUR/$kg_{H_2}$ and 1.7 EUR/$kg_{H_2}$ at 10% p.a. and 5% p.a. WACC respectively. The difference in cost can be partially attributed to the inclusion of the collection infrastructure on the exporter side necessary for transporting the energy carrier from the distributed points of production to the coast for exports.

Concluding the comparison with existing literature yields consistent results where comparison is possible within a broader range. Differences to literature can be tracked and explained based on significantly different technology, cost and model assumptions.

## Conclusion

In this paper we modelled large scale, islanded production and export of chemical energy carriers to Germany (DE) from Australia (AU), Argentina (AR), Denmark (DK), Egypt (EG), Spain (ES), Morocco (MA) and Saudi Arabia (SA). We compared these Energy Supply Chains (ESCs) with their equivalents for domestic production of chemical energy carriers in Germany. For the nine different ESCs we minimised greenfield investment costs to determine the Levelised Cost of Energy (LCoE) per $MWh_{th}$ of the specific chemical energy carriers as well as Levelised Cost of Hydrogen (LCoH) per $kg_{H_2}$ after dehydrogenation of the energy carriers. We determined and used local Renewable Energy Source (RES) potentials via GIS-analysis, considered the influence of local electricity demand on exporters' supply curves and included the intermittency of RES sources via modelled hourly time-series.

In all investigated scenarios domestic sourcing of the individual chemical energy carriers in DE is among the most expensive options. Sourcing through ESCs from other countries is in almost all cases cheaper. Under conservative assumptions of 10% p.a. WACC we find the lowest LCoE (LCoH) for 2030 to be 75 EUR/$MWh_{LHV}$ (2.5 EUR/$kg_{H_2}$) from DK by $H_2$ (g) pipeline. Under optimistic assumptions of 5% p.a. WACC the costs reduce to 50 EUR/$MWh_{LHV}$ (1.7 EUR/$kg_{H_2}$). With technological development, costs further decrease by 2050 to 55 EUR/$MWh_{LHV}$ (10% p.a. WACC) and 36 EUR/$MWh_{LHV}$ (5% p.a. WACC). Other than Denmark, imports from Spain and Western Asia and Northern Africa by hydrogen pipeline are also attractive, e.g. at between 37 EUR/$MWh_{LHV}$ to 42 EUR/$MWh_{LHV}$ (2050, 5% p.a. WACC). With optimistic 5% p.a. WACC assumptions, import options for any to all of the investigated chemical energy carriers and from all exporters become available at cost of 100 EUR/$MWh_{LHV}$

or lower. Importing FT fuels by ship is the most expensive ESC with the largest cost range we investigated, with the costs ranging 90 EUR/MWh$_{LHV}$ to 256 EUR/MWh$_{LHV}$ under best assumptions (2050, 5% p.a. WACC). The lowest cost options and exporters for importing energy are also the lowest cost options for imports of hydrogen. While better financing conditions and fast-track technology development lead to lower costs for imported energy and hydrogen, they do not lead to changes in in the order of preference regarding technology and exporter if they affect all exporters and ESC similarly.

As a rule of thumb LCoE increase with the complexity and energy intensity of an ESC. Costs for RES are the major cost driver usually contributing 39% to 55% to the total cost of an ESC. Electrolysis costs play only a minor role with on average 5% of the costs of an ESC. Costs for ESCs using ammonia, methane, methanol and FT fuels are additionally driven by the costs for synthesis and costs for hydrogen storage using above-ground steel tank, which is used to ensure feedstock availability for the inflexible synthesis processes.

Our findings support the notion that no best one-size-fits-all solution exists for chemical energy carrier imports. In fact no single exporter or ESC shows a unique techno-economic advantage over the others, leading to the conclusion that preference for one energy carrier, ESC and exporter should not be given solely based on small differences in LCoE and LCoH. This allows supply of energy carriers and feedstocks to be sourced from a diverse selection of countries, without affecting costs too strongly. For large volume imports of energy and pure hydrogen, imports by H$_2$ (g) pipeline or electricity by HVDC with domestic electrolysis are favourable. Alternatives incur only slightly higher costs. Our results support the notion that the choice of chemical energy carrier should be based on the requirements for end-use including short-term and long-term storage necessity, energy service conversion efficiency and distribution logistics rather than the cost alone. Accordingly future analysis of full ESCs could provide more insights by using specific demand assumptions and distinguishing between specific energy service or chemical needs. Such analysis could further include local production (captive) scenarios compared to our nation-level analysis. Finally, qualities of ESCs like reliability of supply, long-term cost predictability, local value chains and employment generation, which are generally not represented by the investment costs, need to be investigated and considered.

To overcome the limitations of this study and increase insight into energy imports, future research should focus on differentiating WACC based on exporter and technologies, include spatially resolved RES and collection infrastructure, include additional relevant energy carriers and ESC designs like CO$_2$-cycling and import of secondary, energy-intense products like refined-iron or other valuable hydrocarbons and investigate sensitivities of results also to technical aspects like process flexibilities and synthesis must-run conditions.

## Supporting information

**S1 Appendix. ESF, LCoE and curtailment.**
(PDF)

**S2 Appendix. Technical model structure.**
(PDF)

**S3 Appendix. Model equations.**
(PDF)

**S1 Fig. Energy Supply Chains visualisations.**
(PDF)

**S2 Fig. Electricity generation mix and supply curves.**
(PDF)

**S3 Fig. Cost composition for other years, lower WACC.**
(PDF)

**S4 Fig. RES eligible area masks.**
(PDF)

**S1 Table. Tabular results: Levelised Cost of Energy.**
(PDF)

**S2 Table. Tabular results: Levelised Cost of Hydrogen.**
(PDF)

**S3 Table. Technology assumptions.**
(PDF)

**S4 Table. Conversion efficiencies.**
(PDF)

**S5 Table. Shipping parameters.**
(PDF)

## Acknowledgments

We thank Niclas Mattsson of Chalmers University of Technology, Sweden, for the valuable exchange and help with GlobalEnergyGIS (GEGIS). We also like to express our gratitude to Herib Blanco and the anonymous reviewer, for their time and effort in reviewing our manuscript and providing us with very valuable and helpful comments and suggestions.

## Author Contributions

**Conceptualization:** Johannes Hampp, Tom Brown.

**Funding acquisition:** Michael Düren.

**Methodology:** Johannes Hampp, Tom Brown.

**Supervision:** Tom Brown.

**Writing – original draft:** Johannes Hampp.

**Writing – review & editing:** Michael Düren, Tom Brown.

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
