## [Decision Letter · Decision Letter 0]

13 Jun 2022

PONE-D-22-07863Import options for chemical energy carriers from renewable sources to GermanyPLOS ONE

Dear Dr. Hampp,

Thank you for submitting your manuscript to PLOS ONE. After careful consideration, I feel that it has merit but does not fully meet PLOS ONE’s publication criteria as it currently stands.

In view of the referees’ feedback and my own reading of your paper, I invite you to address all issues noted below, most of which are relatively minor in nature, but nonetheless essential.

In particular, the reviewers request that some small modifications be made that affect different parts of the manuscript but that mainly refer to typographical errors and clarifications on tables, equations and results.

We look forward to receiving your revised manuscript.

Kind regards,

J E. Trinidad Segovia

Section Editor

PLOS ONE

Journal Requirements:

2. We note that Figures 2 and 24 in your submission contain [map/satellite] images which may be copyrighted. All PLOS content is published under the Creative Commons Attribution License (CC BY 4.0), which means that the manuscript, images, and Supporting Information files will be freely available online, and any third party is permitted to access, download, copy, distribute, and use these materials in any way, even commercially, with proper attribution. For these reasons, we cannot publish previously copyrighted maps or satellite images created using proprietary data, such as Google software (Google Maps, Street View, and Earth). For more information, see our copyright guidelines: http://journals.plos.org/plosone/s/licenses-and-copyright.

   a. You may seek permission from the original copyright holder of Figures 2 and 24  to publish the content specifically under the CC BY 4.0 license. 

3. Please ensure that you refer to Figures 12, 13, 14, 15, 16, 17, 18, 19, 20, 21, 22, 23 in your text as, if accepted, production will need this reference to link the reader to the figure.

Reviewers' comments:

Reviewer's Responses to Questions

**Comments to the Author**

1. Is the manuscript technically sound, and do the data support the conclusions?

Reviewer #1: Yes

Reviewer #2: Yes

2. Has the statistical analysis been performed appropriately and rigorously? 

Reviewer #1: Yes

Reviewer #2: N/A

3. Have the authors made all data underlying the findings in their manuscript fully available?

Reviewer #1: Yes

Reviewer #2: Yes

4. Is the manuscript presented in an intelligible fashion and written in standard English?

Reviewer #1: Yes

Reviewer #2: Yes

5. Review Comments to the Author

Reviewer #1: General

- [No action] Great work comparing more carriers than previous studies and with all the data available. My main comments are around the data and sources used which could impact the results and ranking between pathways. Other aspects such as presentation of the results, clarity of the discussion, conclusions supported by the analysis and alike need limited improvement

Literature review

- [1-5] (and many of the studies below) are missing

Design of Energy Supply Chains

- You might consider expanding the scope by one more (steel) and make it "commodities" instead of "chemical energy carriers". You can find the relevant data here [6] and then you can have all the comparison in one place

- Page 4, line 104. Storage can make a big difference for liquid hydrogen since it is expensive to store it in large quantities or to deal with the boil-off. Making the comparison without storage will favor LH2. You can find some data for LH2 storage in [5, 6] for example

Investment optimization problem

- Page 5, line 136. The limits for flexibility of the synthesis units (both ramping rate and minimum turndown) is not yet known and this will constrain the flexibility of the overall plant and determine the sizing of the storage. See [7, 8] for ammonia and [9] for FT

- Page 5, line 144. There are economies of scale that cannot be ignored. You can take some of the scaling exponents from Table 6.1 of [10] and use the standard 0.67 or 0.7 for the remaining technologies. If you take linear scaling, you both overestimate the costs (putting shipping at a disadvantage) and favor liquid hydrogen that requires a larger scale to become cost effective [10]

- Page 5, line 148. If costs scale linearly, how do you prevent that small facilities are built? Do you specify a minimum size below which the route is not possible (see e.g. [11] where they specify minimum trade flows set at 100 kt/yr for long-distance importers, 10 kt/yr for regional importers, and 1 kt/yr for internal EU trade, while in your case it would be a minimum size by technology or overall)

- Page 5, line 149. Fixed efficiencies is not correct for hydrogen liquefaction, electrolysis, ammonia reconversion

for which the values will probably change the most over time. Taking the low values (expected in the long term) already in 2022 would be too optimistic potentially underestimating the cost added by these steps

Choice of technologies and energy carriers

- Page 6, line 174. Alkaline electrolyzers have enough flexibility to provide grid balancing services, so for all practical purposes they are fast enough [12-14]

- Page 6, line 199. Please add in parenthesis the energy efficiency of the LNG liquefaction step (e.g. "liquefaction of methane is energy intensive (96.5%)" - because this energy penalty is much smaller [15]

- Page 7, line 200. Please mention that already today 10% of the global ammonia production (of a total of 183 Mt) is already traded today [10] and this is an advantage that not all the pathways have

- Page 7, line 212. (1) Ref for this statement is missing "high energy requirements (> 25 % of ammonia’s LHV)"; (2) There are a few examples of large-scale application of ammonia cracking (in heavy-water production) [10] and the design would be very similar to SMR furnaces [16] which decreases significantly the risk

- Page 7, line 219. Please add a few sentences on the current scale of methanol reforming and cost ratio with methanol synthesis

- Page 7, line 230. Clarify in this section what the product split is (e.g. to jet fuel/kerosene/diesel). You mention the average LHV, but that is for energy balances purposes. The other relevant aspect is that the products satisfy different end uses and the jet fuel share is more relevant (assuming diesel consumption will decrease significantly once ZEV are adopted)

- Page 7, line 234. (1) For LOHC, one challenge is the small scale today compared to what is needed for global trade. Mention briefly what the assumption is for the costs (same as small scale since costs scale linearly?); (2) Another big limitation is the cost of the carrier itself. Mention in one sentence what the assumption is

- For this section, I realize many of the answers are in Table 7 (and in the Zenodo repository)

- Page 8, line 279. (1) Revise 94.25% based on [9]; (2) Methanol should have a similar flexibility to HB and FT

Countries investigated

- Table 2. Can you eliminate some of the combinations (e.g. HVDC from Argentina and Australia and shipping from Denmark. The former countries are just too far to realistically consider HVDC and shipping from less than 1000 km while having a land connection possible is not realistic either)?

Electricity supply, demand and supply curves

- Page 11, line 345. 1.45 MW/km2 seems extremely low. If I check your ref 25, it seems the 1.45 MW/km2 comes from Section 2.5 that has an arbitrary 1% (145 MW/km2 * 0.01), see e.g. [25] (but there are many others)

- Table 3. The cost reduction profile does not seem to be realistic. Today, global average CAPEX for solar PV is about USD 890/kW [26] and you are assuming it decreases by more than half towards 2030, only to decrease by another 12.5% towards 2040 (vs 2030). I would suggest a higher value for 2030 (and potentially also 2040), while the 2050 value seems reasonable [27] (same comment applies to onshore and offshore wind)

- 10% WACC everywhere does not seem to be the best approach. At least you could use current WACC [28,29]

Results

- Figure 4. LH2 with a similar cost than NH3 by 2030 does not seem right. This might be rooted in the several assumptions that favor LH2 (see previous comments, in particular LH2 storage needs in the terminal). Those should be corrected to give a more complete view

- This is an outcome that should be highlighted more (e.g. conclusions or even abstract) " The exporting country preference order does not change much between the ESCs"

- Figure 5. The almost constant cost decrease towards 2050 for all ESCs is difficult to believe. For instance, one extreme is LH2 where costs for liquefaction, storage, and ships still has to decrease significantly since it is all small scale or pilots today and most of those economies of scale will not be achieved by 2030, so I would expect a larger cost reduction for 2050 (vs 2030). In contrast, for FT, the only step that is missing is electrolysis, FT is commercial and it will be difficult to justify a 50% cost reduction for the synthesis and cost for ships will not decrease as much (as e.g. LH2 ships) over time (other than the use of larger ships which are already the case today)

- Figure 5. I would expect the costs to be much lower since FT efficiency is higher than methanation, the CAPEX is comparable and the transport is much cheaper (so it should be cheaper than CH4 ship). The only downside is the higher CO2 use but that should not be the reason for a 50% higher cost

- Discussion should include (1) How come LH2 is cheaper than CH4 pipeline (i.e. pipeline does not have boil-off losses and it can handle larger flows of energy. It has the losses of the methanation step, but this is only about 20% compared to 30% for liquefaction, so conversion losses are lower and conversion and transport costs are also lower); (2) Why CH4 ship is cheaper than CH4 pipeline (there are losses for liquefaction and the ship has the boil-off losses)

Cost composition and drivers

- Page 17, line 512. From Figure 1, it was not clear that battery storage was included (I guess for increasing the effective capacity factor of PV?). Modify accordingly to make it clear

- Not clear why the renewable electricity cost contribution would be almost double for FT vs CH4 (see comment above) considering the pathway efficiency is similar. A higher flexibility for FT (or more comparable to methanation) should also make hydrogen storage cost for FT smaller

Hydrogen import costs for 2030 to 2050

- Please make more explicit and clearer earlier in the text that you are covering energy transport (with different carriers as end use) and hydrogen transport (all reconverted to hydrogen). Up to this point, I still had the question if for instance FT was being used as hydrogen carrier (which would make little sense) but I realize now the figures above are just comparing different carriers delivered (i.e. not directly comparable)

- Page 18, line 543. This point should make it clearly to the conclusions "by 2050 as most exporting countries except for AR, AU and DE show nearly identical LCoHs", so where it comes from does not really make a difference

Lower WACC

- More important than testing lower WACC across all countries is considering the WACC differentials across countries. While there are no good values for 2050, the most sensible assumption would be to assume they stay similar to the current values. This would be a better sensitivity (or base) scenario since the only aspect you are measuring by doing it all 5% instead of 10% is how much cheaper it becomes which was already evident from Figure 8. I see this is in the Limitations section but I am not sure if it is good enough

- Page 21, line 601. Can you somehow include the comparison with current commodity prices in one of the figures above (e.g. an additional dot with a different marker or a shaded area for each commodity)? This will make it easier for the reader that might browse through the figures (instead of reading the entire text)

- Page 21, line 618. Make a figure with the CO2 costs

Conclusions

- Page 23, line 731. This is the part that is not fully self-explanatory, when you mention/use LCOE, it is not 100% clear that it means comparing pathways with different molecules at the end, perhaps "Levelised Costs of Energy (LCoEs) per MWhth with different molecules as delivered products"? Feel free to choose a different wording but it needs to be clearer

- Make sure the comments above where I mention that those ideas should make it to the conclusions are incorporated

Table 7

- Pipeline cost should be a function of capacity. It is not the same a small pipeline than a large one. Gas pipelines are cheaper than hydrogen [17-19]

- Why not considering SMR with CCS? It is a relatively small cost penalty (USD 70-100/tCO2) for negative emissions (since that CO2 originally comes from DAC and I am assuming this is for SNG reconversion to hydrogen at the importing terminal)

- Depending on how much influence the CO2 liquefaction has, you might want to use some more pessimistic assumptions, [20] has almost double the cost and energy consumption

- The CAPEX for dehydrogenation in [Runge 2020] is too high because it is meant to be for HRS (i.e. small scale), it is more in the USD 100-250/kW range [10]

- HVDC. Please include complete reference

- HB. The CAPEX seems to be highly overestimated, please double check the value and see [7,8,10,20]

- Hydrogen regasification. The CAPEX seems to be too low, see e.g. [4, 22]

- Hydrogen storage tank. Specify pressure

- ASU. Since the conversion seems to have gone wrong for HB (from [Morgan 2013]), please double check the CAPEX because it seems too high (see [7,8,10,20])

- Methane fill compressor station. Why would this be needed? (the name seems to indicate compression for CNG cars or similar. If you mean a standard compression station for pipeline transport, then delete the "fill")

- Hydrogen pipeline. Specify the diameter assumption for the CAPEX

- Methane liquefaction. It seems something went wrong for the CAPEX (0 EUR/kW for 2040 and 2050). I would expect somewhere in the 400-600 EUR/kW range [23]

- FT synthesis. (1) Unclear why the learning would be about 20% cost reduction for methanation and about 50% for FT (I would argue it should be similar to methanation since FT is already a commercial technology and there are limited improvements to reach 50% cost reduction); (2) EUR 1600/kW seems too high [24]

- Ammonia cracker. EUR 1400/kW is too high, see Figure 2.13 of [10]. You might as well consider that in the medium term (2030) relatively small crackers are used and the full economies of scale are reaped by 2040 and beyond

- Hydrogen liquefaction. Benchmark CAPEX with [10] and there should probably be a cost decrease over time

Table 8

- FT-CO2. Can you change the basis from 1t of fuel to 0.56 MWh of fuel (which the rows below seem to use and this would allow for a direct comparison)

[1] https://www.ewi.uni-koeln.de/en/publications/estimating-long-term-global-supply-costs-for-low-carbon-hydrogen/

[2] https://www.globh2e.org.au/_files/ugd/8d2898_2778fbb92232442ba17942dc47b5f845.pdf

[3] https://www.sciencedirect.com/science/article/abs/pii/S0360319921016815

[4] https://doi.org/10.1016/j.ijhydene.2020.09.017

[5] https://repository.tudelft.nl/islandora/object/uuid%3Ad2429b05-1881-4e42-9bb3-ed604bc15255

[6] https://doi.org/10.1016/j.enconman.2022.115268

[7] DOI: 10.1039/d0ee01707h

[8] https://doi.org/10.1016/j.ijhydene.2019.11.028

[9] https://doi.org/10.1021/acs.est.0c07955

[10] IRENA report. Global hydrogen trade to meet the 1.5C climate goal: Part II - Technology review of hydrogen carriers

[11] https://www.belfercenter.org/sites/default/files/files/publication/Report_EU%20Hydrogen_FINAL.pdf

[12] https://www.nrel.gov/docs/fy14osti/61758.pdf

[13] https://ietresearch.onlinelibrary.wiley.com/doi/10.1049/iet-rpg.2020.0453 (and ref 17 therein)

[14] https://www.thyssenkrupp.com/en/newsroom/press-releases/pressdetailpage/thyssenkrupps-water-electrolysis-technology-qualified-as-primary-control-reserve--eon-and-thyssenkrupp-bring-hydrogen-production-to-the-electricity-market-83355

[15] https://pubs.rsc.org/en/content/articlelanding/2020/se/d0se00067a

[16] https://www.sciencedirect.com/science/article/abs/pii/S0306261920314549

[17] http://dx.doi.org/10.1016/j.ijhydene.2015.06.090

[18] doi:10.1016/j.ijggc.2011.09.008

[19] https://gasforclimate2050.eu/?smd_process_download=1&download_id=471

[20] http://www.co2europipe.eu/

[21] https://doi.org/10.1016/j.ijhydene.2018.06.121

[22] https://gasforclimate2050.eu/?smd_process_download=1&download_id=718

[23] https://doi.org/10.1016/j.rser.2018.09.027

[24] https://doi.org/10.1016/j.rser.2017.05.288

[25] https://doi.org/10.1109/JPHOTOV.2021.3136805

[26] https://www.irena.org/publications/2021/Jun/Renewable-Power-Costs-in-2020

[27] https://doi.org/10.1016/j.esr.2021.100636

[28] http://aures2project.eu/wp-content/uploads/2021/06/AURES_II_D5_2_financing_conditions.pdf

[29] https://doi.org/10.1016/j.eneco.2020.104783

Reviewer #2: The article investigates synthetic chemical energy carriers, hydrogen, methane, methanol, ammonia and Fischer-Tropsch fuels, produced using electricity from Renewable Energy Source (RES) as fossil substitutes. They model the sourcing of feedstock chemicals, synthesis and transport along nine different Energy Supply Chains to Germany and compare import options for seven locations around the world against each other and with domestically sourced alternatives on the basis of their respective cost per unit of hydrogen and energy delivered. The article is rigorous and methodological choices are well motivated. I recommend publication after minor revisions.

It is great that you make the model openly available but it would also good if you could include some or the equations used in the model also in the main article so it would be easier to follow for the reader.

Please explain p.a., I assume it stands for per year.

Please write out WACC where you first introduce it, it is explained later in the text.

You use 5% p.a. in abstract but assume 10% in the main results why highlight the 5% in the abstract? I think this needs to be motivated. What do you think is the best value to use?

Line 427 You write “The lowest cost option for import is at 77EUR/MWhLHV by H2 pipeline from DK. Runner-up ESCs are also by H2 pipelines and imports by HVDC from DK at 92EUR/MWhLHV.” You mention H2 pipelines from DK in both pathways is it by mistake? I do not fully follow.

You do consider expected domestic use of the chemical feedstock. Would it for the remote location potentially also make a difference if considering the need from countries that are closer than Germany? How would this look at a global scale?

How you take the synthesis process flexibility into account seem good but it would be good if you could describe how it is incorporated in the model a bit more. “In addition to the economic perspective of operating synthesis processes at a high utilisation rate, some synthesis processes may be designed for continuous operation from a chemical process point of view [16, 19] and not be suited for flexible operation or standby. We take this into account be setting the must-run capacity of the methanation and ammonia synthesis processes to 30% each, based on what could potentially be feasible for the methanation [16] and Haber-Bosch [19] processes. Methanol and FT fuel synthesis are assumed to run at a minimum of 94.25% capacity corresponding to a maximum of 3 weeks downtime for e.g. maintenance per year.”

From your figures, I can see that you are considering hydrogen storage. I would be good if you could elaborate a bit more on how you have determined the needed size of the hydrogen storage for the different chemical feedstock. I have not managed to find this in the text, but apologies if it is there and I missed it.

For the direct air capture you are referring to the wrong technology data sheet (technology_

data_for_industrial_process_heat_0002.xlsx). Should be that connected to carbon capture and storage I think.

It is not clear for what demand of end energy in German the cost is calculated. I assume the larger the demand the higher the demand due to the increased cost for electricity.

The argument for why you consider DAC and not recirculating CO2 could be further elaborating on and backed up with references.

The cost difference between e-methanol and ammonia from you model is not seen in al studies for example not in Korberg et al (2021). It would be great if you could explain the reason for the significantly higher cost of methanol.

Consider changing the name of the y-axis on Figure 3 “Marginal electricity cost”. For some it may be confusing with the term marginal as this is used for different things. I am not sure I fully understood how it is derived but is the word marginal needed.

References

Korberg, A., Brynolf, S., Grahn, M., Skrov, I., 2021. Techno-economic assessment of advanced fuels and propulsion systems in future fossil-free ships. Renewable and Sustainable Energy Reviews 142.

6. PLOS authors have the option to publish the peer review history of their article (what does this mean?). If published, this will include your full peer review and any attached files.

Reviewer #1: **Yes: **Herib Blanco

Reviewer #2: No

---

## [Author Response · Author response to Decision Letter 0]

29 Dec 2022

Please find all our responses detailed in the dedicated "response to reviewers" file which we have uploaded to the Editorial Manager.

---

## [Decision Letter · Decision Letter 1]

23 Jan 2023

Import options for chemical energy carriers from renewable sources to Germany

PONE-D-22-07863R1

Dear Dr. Hampp,

We’re pleased to inform you that your manuscript has been judged scientifically suitable for publication and will be formally accepted for publication once it meets all outstanding technical requirements.

Kind regards,

J E. Trinidad Segovia

Section Editor

PLOS ONE

Additional Editor Comments (optional):

Reviewers' comments:

Reviewer's Responses to Questions

**Comments to the Author**

1. If the authors have adequately addressed your comments raised in a previous round of review and you feel that this manuscript is now acceptable for publication, you may indicate that here to bypass the “Comments to the Author” section, enter your conflict of interest statement in the “Confidential to Editor” section, and submit your "Accept" recommendation.

Reviewer #2: All comments have been addressed

2. Is the manuscript technically sound, and do the data support the conclusions?

Reviewer #2: Yes

3. Has the statistical analysis been performed appropriately and rigorously? 

Reviewer #2: N/A

4. Have the authors made all data underlying the findings in their manuscript fully available?

Reviewer #2: Yes

5. Is the manuscript presented in an intelligible fashion and written in standard English?

Reviewer #2: Yes

6. Review Comments to the Author

Reviewer #2: (No Response)

7. PLOS authors have the option to publish the peer review history of their article (what does this mean?). If published, this will include your full peer review and any attached files.

Reviewer #2: No

---

## [Editor Report · Acceptance letter]

30 Jan 2023

PONE-D-22-07863R1 

Import options for chemical energy carriers from renewable sources to Germany 

Dear Dr. Hampp:

I'm pleased to inform you that your manuscript has been deemed suitable for publication in PLOS ONE. Congratulations! Your manuscript is now with our production department. 

Kind regards, 

on behalf of

Dr. J E. Trinidad Segovia 

Section Editor

PLOS ONE